# Structural basis for self-discrimination by neoantigen-specific TCRs

John P. Finnigan[1,2,3,4,10], Jenna H. Newman[1,2,3,10], Yury Patskovsky[5,6,10], Larysa Patskovska[5,6], Andrew S. Ishizuka[7,8], Geoffrey M. Lynn[7,8], Robert A. Seder ⓘ[7], Michelle Krogsgaard ⓘ[5,6] ✉ & Nina Bhardwaj ⓘ[1,2,3,9] ✉

T cell receptors (TCR) are pivotal in mediating tumour cell cytolysis via recognition of mutation-derived tumour neoantigens (neoAgs) presented by major histocompatibility class-I (MHC-I). Understanding the factors governing the emergence of neoAg from somatic mutations is a major focus of current research. However, the structural and cellular determinants controlling TCR recognition of neoAgs remain poorly understood. This study describes the multi-level analysis of a model neoAg from the B16F10 murine melanoma, H2-$D^b$/Hsf2 p.K72N$_{68-76}$, as well as its cognate TCR 47BE7. Through cellular, molecular and structural studies we demonstrate that the p.K72N mutation enhances H2-$D^b$ binding, thereby improving cell surface presentation and stabilizing the TCR 47BE7 epitope. Furthermore, TCR 47BE7 exhibited high functional avidity and selectivity, attributable to a broad, stringent, binding interface enabling recognition of native B16F10 despite low antigen density. Our findings provide insight into the generation of anchor-residue modified neoAg, and emphasize the value of molecular and structural investigations of neoAg in diverse MHC-I contexts for advancing the understanding of neoAg immunogenicity.

The T cell receptor (TCR) is a variable heterodimeric protein complex that non-covalently binds to the surface-bound peptide-major histocompatibility complex (pMHC), which presents peptide antigens derived from degraded intracellular proteins[1]. Anti-tumour T cell immunity is mediated by the physical interaction between T cell receptors (TCR) and tumour antigens presented by pMHC on tumour cells[2]. Tumour cells accumulate somatic non-synonymous mutations encoding variant proteins that ultimately degrade to form mutation-derived tumour neoantigens (neoAg)[3]. Analogous to pathogens, tumours evolve in hosts under selective pressure from endogenous and treatment-induced immunity[4]. However,

immunogenic neoAg can persist despite selective immunoediting and are increasingly recognised as the primary target of tumour-reactive TCRs[5–7]. There are now multiple clinical trials associating neoAg-reactive T cells with positive clinical outcomes (for example, radiographic regression of established tumours and/or prolonged disease-free and overall survival) for patients treated with therapeutic vaccines[8–14] cell-based therapies[15–17], and immune checkpoint blockade[18–22]. However, because of historical difficulties associated with prospectively studying clinically relevant human neoAg-reactive TCRs, only a fraction of the TCRs identified to date have received detailed in vitro and in vivo characterisation[15,23–25].

[1]Icahn School of Medicine at Mount Sinai, One Gustave L. Levy Pl., New York, NY, USA. [2]Tisch Cancer Institute, Icahn School of Medicine at Mount Sinai, 1470 Madison Ave., New York, NY, USA. [3]Department of Medicine, Division of Hematology and Medical Oncology, Mount Sinai Hospital, New York, NY, USA. [4]Department of Surgery, Division of Thoracic and Cardiac Surgery, Brigham and Women's Hospital, 75 Francis St., Boston, MA, USA. [5]Department of Pathology, New York University Grossman School of Medicine, New York, NY, USA. [6]Laura and Isaac Perlmutter Cancer Center at NYU Langone Health, New York, NY, USA. [7]Vaccine Research Center, National Institute of Allergy and Infectious Diseases, National Institutes of Health, Bethesda, MD, USA. [8]Barinthus Biotherapeutics, Germantown, MD, USA. [9]Parker Institute for Cancer Immunotherapy, Francisco, CA, USA. [10]These authors contributed equally: John P. Finnigan, Jenna H. Newman, Yury Patskovsky. ✉e-mail: michelle.krogsgaard@nyulangone.org; nina.bhardwaj@mssm.edu

High functional avidity/structural affinity has emerged as a recurrent feature of neoAg-reactive TCRs[26] and may be necessary to recognise tumour cells naturally selected for low target antigen surface density. In early examples, this has been shown to derive from TCR recognition of structural differences between mutation-derived neoAg peptide and the corresponding wild-type (WT) peptide[27–29], but the broader generalizability of these findings remains unknown. Many other core questions remain unanswered, such as why non-synonymous mutations are rarely recognised by TCRs; and how some neoAg-reactive TCRs selectively recognise mutated peptides and do not cross-react with the corresponding wild-type peptides, whereas others exhibit significant cross-reactivity[27–30]. Structure-guided mechanistic answers to these questions might enable the prediction of neoAg-reactive TCR activity as well as potential toxicities resulting from cross-reactivity, potentially enabling the rapid translation of safe and effective neoAg-reactive TCRs into the clinic.

To systematically address these questions pertaining to neoAg-reactive TCRs, we employed the B16F10 murine melanoma cell line, an orthotopic implantable tumour model syngeneic to C57BL/6 mice that exhibits limited spontaneous immunogenicity and is refractory to multiple types of immunotherapy, including checkpoint blockade[31]. We reasoned that neoAgs identified in this model would approximate neoAgs observed in advanced human cancers more closely than exogenous model antigens such as ovalbumin[32], thereby improving the biological relevance of our findings. Furthermore, there are several well-known conventional tumour-associated antigens (gp100, Trp2, Tyrp1) relevant in the B16F10 model which can serve as comparators for functional and structural studies. These tumour-associated antigens (TAAs) often do not elicit a robust or exclusively tumour-specific endogenous anti-tumour immune response and corresponding tumour growth control[33,34], underscoring the need to identify and study neoAgs in the B16F10 model.

Here we perform whole exome and transcriptomic sequencing of B16F10 and characterise a subset of expressed non-synonymous mutations via in vivo validation. We immunise mice with synthetic peptides corresponding to selected mutations and characterise the vaccine-induced CD8+ T cell response to seven neoAgs from over 50 predicted neoAgs. We isolate, clone, and perform functional analyses of cognate TCRs recognizing each neoAg. Among them, only the TCR targeting H2-D$^b$-restricted Hsf2 p.K72N ('p' indicating peptide residue) confers specific recognition of the B16F10 cells in vitro and demonstrates anti-tumour effect in vivo, albeit dependent upon sufficient tumour expression of neoAg Hsf2 p.K72N. Finally, using biochemical and cellular assays in combination with high-resolution crystal structures of the neoAg Hsf2 p.K72N-H2-D$^b$ complex, with and without a corresponding reactive TCR, we determine the structural requirements for TCR antigen recognition and selectivity. We observe that Hsf2 p.K72N is discriminated by both the MHC and cognate TCR from the WT Hsf2. We determine that Hsf2 p.K72N is a group II neoAg with a mutation at an anchor residue. Group II neoAg are typically minimally cross-reactive with their corresponding WT peptides due to discrimination at the MHC level and thereby may resemble non-self epitopes generated in the course of viral or bacterial infections[35].

## Results

### Identification of neoantigens in B16F10 melanoma

To identify B16F10 neoAgs, we performed paired exome sequencing of cultured B16F10 murine melanoma tumour cells and reference C57BL/6 splenocytes, as well as bulk RNASeq analysis of resected B16F10 tumours (Fig. 1a). Variant expression was quantified by local assembly and allele-specific quantification of mutated and reference transcripts. The peptide-MHC-I binding prediction tool NetMHCpan (v.4.1) was then used to identify candidate neoAg for further study, in accordance with published methods[36,37]. We then performed murine immunization studies using SNAPvax™, a peptide-based vaccine that is

conjugated to an adjuvant small molecule imidazoquinone-based Toll-like Receptor 7/8 agonist (TLR7/8a) and contains charge-modifying groups to accommodate a wide variety of peptide chemistries; this cancer vaccine platform has yielded robust anti-tumour T cell immunity enabled by enhancements in dendritic cell recruitment and antigen uptake, which in turn reduced tumour growth in multiple models[38,39]. For these initial immunization studies, we developed SNAPvax™ formulations incorporating twelve distinct 25mer ("long") neoantigenic peptides (Supplementary Table 1). To minimise the risk of antigenic competition, screening immunization was performed individually with one peptide antigen specificity, for each of the 12 tested neoAg, and three control non-mutated tumour-associated antigens (TAA)[40–42]. We observed both vaccine-elicited neoAg-specific CD4+ and CD8+ T cells amongst splenocytes for 4 and 7 neoAgs, respectively, of nearly 50 long peptide-derived MHC-I-restricted neoAg peptides surveyed (Fig. 1b, c, Supplementary Figs. 1, 2), as defined by robust IFNγ production by T cells in response to stimulation with neoantigenic peptides and, for the CD8+ T cell compartment specifically, tetramer staining of T cells with MHC-I tetramers. To determine the minimal peptide epitope for the 7 MHC-I-restricted neoAg hits, we immunised mice with various minimal epitopes derived from the long peptide vaccine formulations and assessed neoAg-reactive T cell yield via tetramer staining, akin to that performed in Fig. 1b (Fig. 1d). A summary of these immunogenic MHC-I-restricted epitopes, as well as four non-mutated previously characterised tumour antigens[40–42], and their predicted binding affinities of wild type (WT) versus mutant (MT) peptide to MHC is shown; these predictions revealed that pMHCs in our system span the spectrum of mutant peptide-MHC affinity and specificity for mutant peptide (ratio of mutant to wild type affinity for MHC) (Fig. 1e, f). Altogether, we observed that predicted neoAgs are indeed immunogenic in vivo and determined the minimal epitopes that elicit T cell immunity for seven neoAgs, enabling further characterization.

### Neoantigen-reactive CD8+ T cells recognise cognate pMHC

Next, we sought to identify neoAg specific-TCRs and engineer neoAg-reactive T cells for mechanistic analysis of TCR-pMHC interactions. To ensure a consistent, clonal, population of TCRs in all subsequent studies, we first single-cell sorted pMHC tetramer+ T cells elicited from vaccination, as described in methods, and performed 5' rapid amplification of cDNA ends (RACE) and sequencing of the TCR alpha (TCRα) and TCR beta (TCRβ) variable chains. We successfully sequenced and reconstructed paired TCRα and TCRβ. Knowing the TCRα and TCRβ sequences, we cloned nine identified neoAg-reactive TCRs into murine stem cell virus (MSCV) plasmid vectors, as well as four non-mutated tumour antigen-specific TCRs for comparison (Fig. 2a). Then, using either vaccine-elicited or retrovirus-transduced TCR-transgenic (tgTCR) CD8+ T cells, we confirmed antigen-induced cytokine production for all identified TCRs (Fig. 2b, c, Supplementary Figs. 3a, b, 4a). Importantly, we verified that all tgTCR CD8+ T cells expressed similar TCR surface levels (Supplementary Fig. 3c, Supplementary Fig. 4b). We then assessed antigen-induced cytokine production by all identified TCRs in response to the mutant (MT) and wild type (WT) peptide (Fig. 2c, d). Of particular interest were T cells harbouring the tgTCR 47BE7, which targets the H2-D$^b$-restricted neoAg Heat Shock Protein 2 (Hsf2 p.K72N$_{68-76}$). TCR 47BE7 (V$_\alpha$7-1:J$_\alpha$21, V$_\beta$2:D$_\beta$2:J$_\beta$2-1) is derived from a vaccine-induced cytotoxic CD8+ T cell clone that recognises the H2-D$^b$/Hsf2 p.K72N$_{68-76}$ with sub-nanomolar functional avidity (EC$_{50}$ 5.61 pM) (Fig. 2d). The TCRs exhibited variable selectivity for their cognate neoAg ranging from complete specificity (29BF8, 44CH2) to complete cross-reactivity (46AD8, 50AD1) (Fig. 2d). We then assessed TCR recognition of B16F10 target cells. Notably, only TCR 47BE7 exhibited T cell effector function upon co-culture with unmodified B16F10 cells

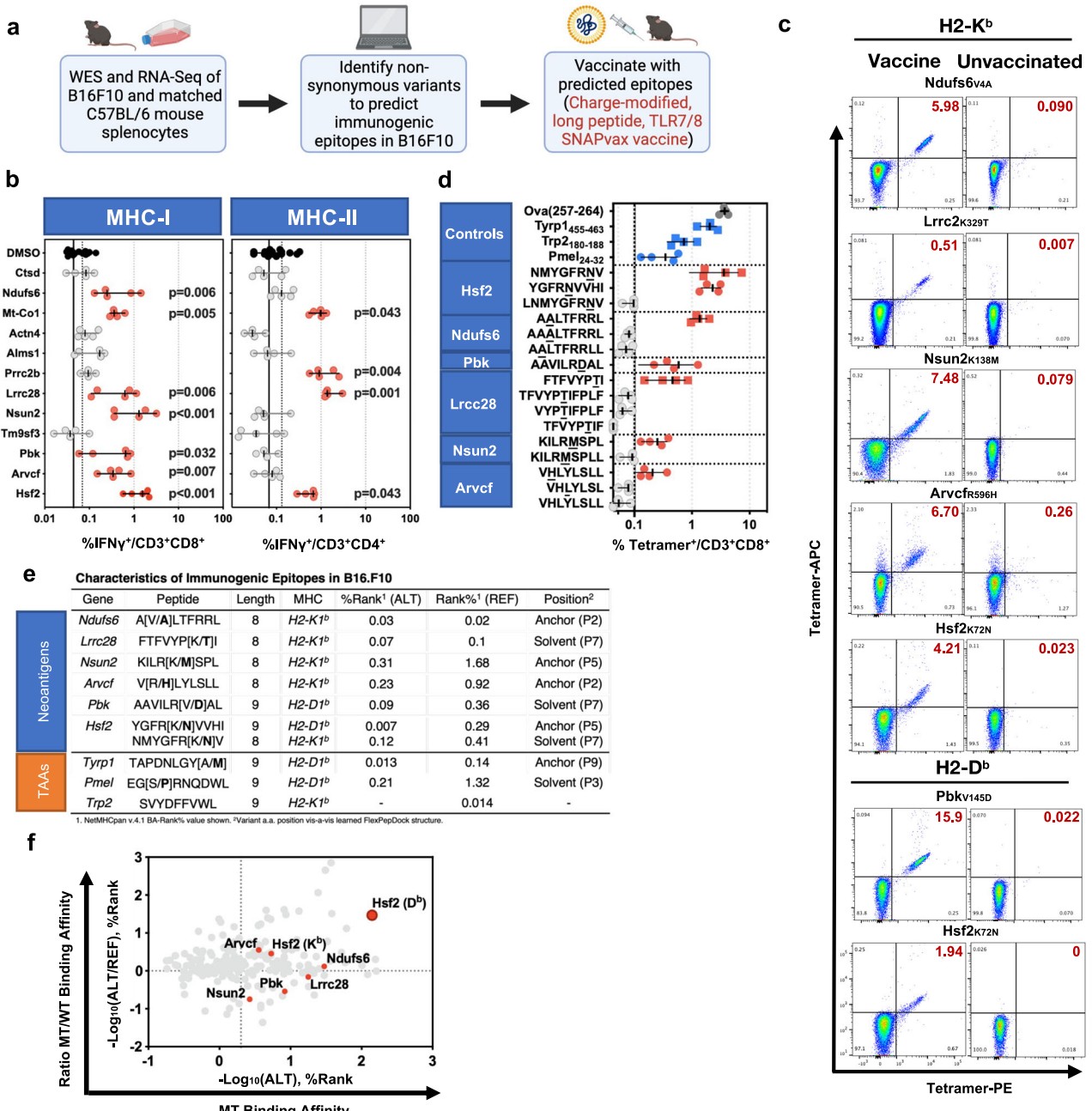

**Fig. 1 | Identification of neoantigenic epitopes in the B16F10 melanoma model. a** Schematic (made using Biorender with full license) depicting B16F10 neoantigen identification. **b** C57BL/6 mice (*n* = 5 independent biologic replicates) were immunised with peptide vaccine targeting putative B16F10 neoantigens. Seven days post-immunization splenocyte-derived T cells were stimulated with mutant peptide (solubilised in DMSO) for 6 h then IFNγ production was measured by flow cytometry. Symbol indicates individual mice (*n* = 5/condition), error bars indicate the group median, ±95% confidence interval (CI). Solid line indicates assay lower limit of detection (LLD). Dashed indicates upper limit of 95% CI for negative responses. Statistical analysis: one-way Kruskal–Wallis test, followed by Dunn's test for multiple comparisons, with alpha level set to 0.05. Colour key: red; T cell-elicited response significantly above the LLD, grey; insignificant T cell response, black; T cells not stimulated with any peptide (DMSO only control). Where *p* < 0.001, *p* value was too low for Prism software to provide an exact value. **c** C57BL/6 mice (*n* = 4/group, repeated four times) were immunised with (long) peptide vaccine targeting putative B16F10 neoantigens. Seven (7) days post-

immunization, flow cytometry was performed on splenocyte-derived CD8+ T cells. Flow cytometry plots are organized in columns; data from vaccinated mice (left) vs unvaccinated (right). Representative tetramer staining is shown. Gating strategy shown in Supplementary Fig. 2. **d** As in (**b**). Error bars depict the group median, ±95% confidence interval (CI); *n* = 5 independent biologic replicates. Peptide stimulation of T cells by tumour-associated antigens (TAAs) [blue], (non-tumour) OVA antigen [dark grey], neoantigens eliciting measurable tetramer response [red], neoantigens not eliciting measurable tetramer response [light grey]. **e** Attributes of neoantigens (top) predicted computationally and TAAs previously characterised[40–42]. Binding affinity (BA)-Rank values are shown for mutated (ALT) and wild type (REF) peptides. Table subscript legend is as follows: [1]NetMHCpan v.4.1 BA-Rank% value. [2]Variant a.a. position vis-à-vis learned FlexPepDock structure. **f** In silico MHC-I binding affinity analysis of Hsf2 p.K72N$_{68-76}$ demonstrates high affinity binding and high differential binding affinity between ALT and REF peptides, hereafter referred to as mutant (MT) and wild type (WT) peptides, respectively. Source data are provided as a Source Data file.

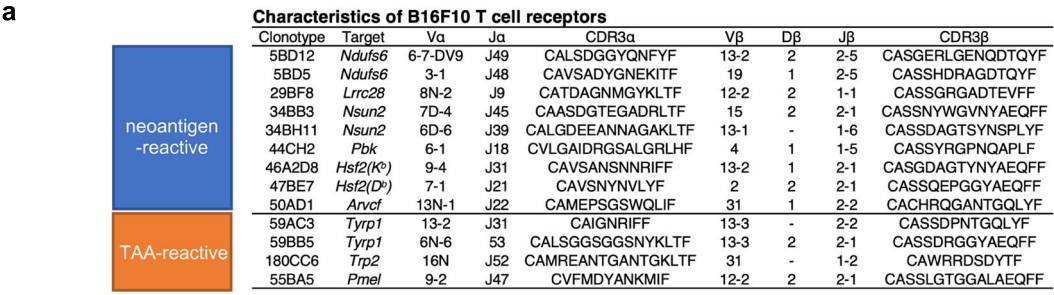

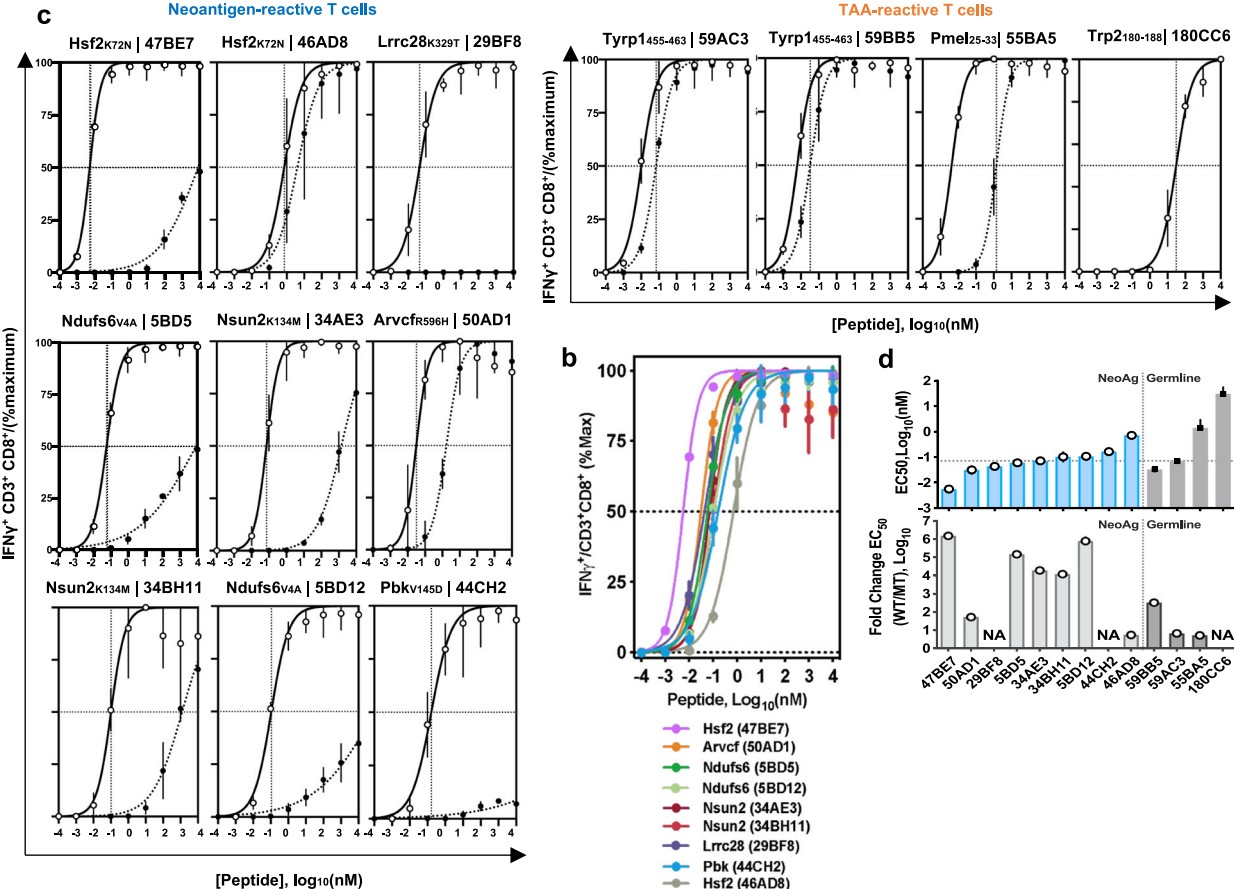

**Fig. 2 | Neoantigen-reactive CD8+ T cells recognise cognate peptide in vitro.**
**a** Table detailing attributes of tumour-reactive T cell receptors (TCRs), including neoantigen-reactive (top) and tumour-associated antigen-reactive (TAA-reactive) (bottom) TCRs isolated upon vaccination. **b** Neoantigen-reactive CD8+ T cell responses were induced by vaccination; isolated tetramer+ T cells were stimulated for 6 h with (varying concentrations of) cognate peptide and αCD28 and subsequently analysed via intracellular staining (ICS) flow cytometry for IFNγ expression. **c** Transgenic (tg)TCR CD8+ T cells were co-incubated with varying concentrations of cognate mutant (MT) peptide or wild type (WT) peptide (x-axis) and αCD28 for 6 h; intracellular staining (ICS) was performed subsequently. Percentage of T cells

expressing IFNγ of a parent CD3+CD8+ population is shown on y-axis, normalised to maximum IFNγ expression. MT peptide values are shown as clear circles, and WT as filled, black circles. Trp2-reactive TCR 180CC6 only recognises a WT peptide (shown with clear circle). **d** Neoantigen-reactive TCR half-maximal (EC$_{50}$) cytokine production concentration (top). TgTCR CD8+ T cells expressing the indicated TCR were stimulated, as described in (**c**), with titrated mutant (MT) or wild-type (WT) peptide and IFNγ production was measured by ICS. Symbols indicate median of biologic replicates (n = 3/condition), ±95% CI. Dashed horizontal line indicates mean half-maximal response (EC$_{50}$) for tested neoAg TCR. (Bottom) Ratio of WT/MT EC$_{50}$ as log$_{10}$ fold change. Source data are provided as a Source Data file.

in vitro (Fig. 3a). This was in notable contrast to control TCRs targeting tumour-associated antigens (e.g., gp100, Trp2, Tyrp1), which typically elicited activity with exposure to unmodified B16F10 cells in vitro (Fig. 3a, Supplementary Fig. 5). We hypothesised that this was due to heterogenous, and comparatively low transcript expression of all tested neoAg, which ranged from hundreds- to thousands-fold lower compared to that of tested tumour-associated antigens (Supplementary Fig. 3d). Consistent with this hypothesis we found that overexpression of all tested neoAg in B16F10, with the exception of Lrrc28 and Nsun2, yielded improved

recognition by tgTCR+ CD8+ T cells in vitro (Fig. 3a–c). As expected, the 47BE7 TCR+ CD8+ T cells exhibited even higher levels of IFNγ upon overexpression of the Hsf2 neoAg by B16F10 (Fig. 3c). In vivo, therapeutic immunization with the minimal epitope Hsf2 p.K72N elicited an enrichment of 47BE7+ CD8+ T cells amongst tumour-infiltrating lymphocytes (TILs) (Supplementary Fig. 6a) and delayed tumour growth (Fig. 4a, b). Adoptive cell transfer (ACT) of tgTCR 47BE7+CD8+ T cells also delayed B16F10 tumour growth upon overexpression of Hsf2 p.K72N in the B16F10 tumour line (Fig. 4c, d, Supplementary Fig. 6b, c). Given demonstrated sensitivity,

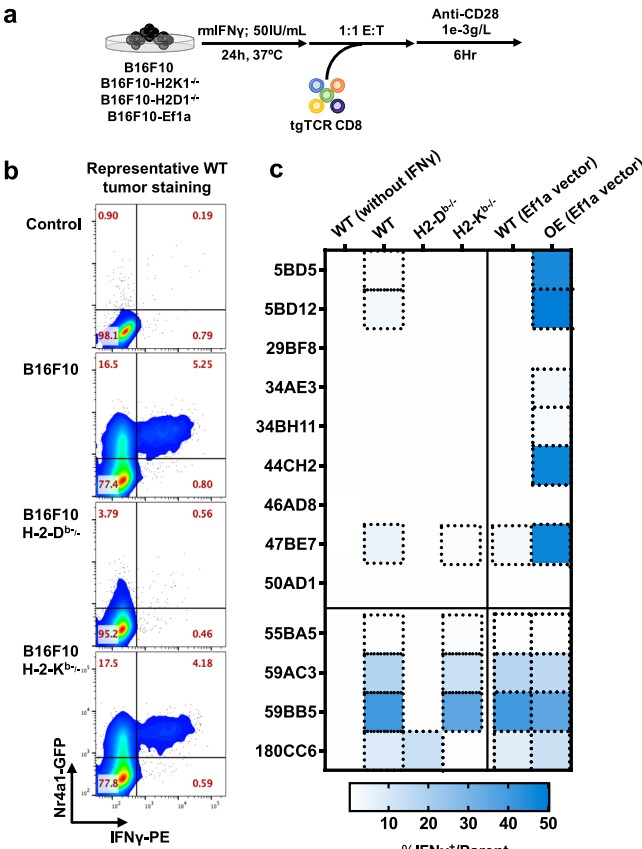

**Fig. 3 | Hsf2 neoantigen-reactive CD8⁺ T cells recognise tumour cells in vitro. a** Wild type B16F10 (WT), B16F10 lacking either MHC-I H2-D$^b$ or H2-K$^b$ (B16F10-H2D$^{b-/-}$ and B16F10 H2K$^{b-/-}$, respectively) or B16F10-Ef1a (overexpressing neoantigenic or TAA peptide) were plated, exposed to recombinant murine IFNγ (rmIFNγ), and T cells expressing tgTCRs engineered from Nr4a1-eGFP mice were added at a 1:1 effector:target (E:T) ratio and co-incubated. Nr4a1-GFP is a marker of TCR signal transduction. **b** Cytokine production was measured by intracellular flow cytometry (*n* = 3 biological replicates/condition, repeated 3 times). Representative flow cytometric analysis of CD8⁺TCR-47BE7⁺ (Hsf2-reactive) cells exposed to (WT) B16F10 target cells is shown. Gating strategy for flow cytometry is shown in Supplementary Fig. 5. **c** Heatmap shown summarises the frequency of IFNγ⁺ cells of the parent CD8⁺tgTCR⁺ population. TCR clone names are shown for each row. IFNγ was withheld from conditions shown in the leftmost heatmap column (indicated as 'WT (without IFNγ)' to serve as a negative control. 'WT (Ef1a vector)' indicates that B16F10 was transduced with an "empty antigen" Ef1a lentiviral construct, meanwhile 'OE (Ef1a vector)' indicates that B16F10 was transduced (for antigen overexpression [OE]) with the appropriate antigen matching the TCR indicated.

selectivity, and functional activity, the Hsf2 p.K72N-reactive TCR was selected for further characterization.

**Stabilisation of pMHC by mutant-specific anchor residue**

Previous studies have shown that the position of the mutated amino acid with respect to peptide length can be used to organise neoAgs into two principal groups[43,44]. Namely, neoAgs in which the mutated amino acid side-chain is solvent facing and may form inter-molecular bonds with incoming TCR directly (group I); versus neoAgs in which the mutated amino acid side-chain is not solvent facing, and instead buried within and interacts predominately with the MHC-I binding pocket (group II).

Hsf2 p.K72N$_{68-76}$ is a H2-D$^b$-restricted mutated peptide ($_{68}$YGFR**N**VVHI$_{76}$) derived from Heat shock factor 2 (Hsf2, Uniprot: P38533). The underlying point mutation results in substitution of a basic lysine (Lys/K) residue at position 5 of WT Hsf2$_{68-76}$ (pK$_5$; 'p' indicating

peptide residue, with the number designating the position of the residue in the peptide starting from the N-terminus) for a polar non-charged residue asparagine (Asn/N (pN$_5$)). In silico binding analysis with NetMHCPan4.1 predicted that both Hsf2 p.K72N$_{68-76}$ [0.007%Rank] and WT Hsf2$_{68-76}$ [0.29%Rank] can bind H2-D$^b$, albeit with a significantly different affinity (Fig. 1f). To confirm the observed difference in binding affinity we performed cell-based RMA-S in vitro binding assays, which showed half-maximal stabilisation of cell surface H2-D$^b$ (EC$_{50}$) by Hsf2 p.K72N$_{68-76}$ at 4.985 nM that was comparable to that of control agonist peptide LCMV gp$_{33-41}$ 4.38 nM, and significantly lower than that of Hsf2$_{68-76}$ 883 nM (Fig. 5a). Based on this observation we hypothesised that H2-D$^b$/Hsf2 p.K72N is a murine prototype group II neoAg[35], in which immunogenicity is at least in part derived from improved binding of the MT peptide with H2-D$^b$, likely secondary to the mutated residue pN$_5$. Furthermore, given its high predicted MHC-I binding affinity for MT peptide, with a large differential in binding affinity between the MT and WT peptide, we hypothesised that detailed structural characterization of the H2-D$^b$/Hsf2 p.K72N$_{68-76}$ complex might elucidate principles governing the immunogenicity of group II neoAgs.

Analysis of published H2-D$^b$ crystal structures demonstrated a conserved peptide binding mode mediated by hydrophobic interactions between conserved residues lining the H2-D$^b$ A-B-D and F-pockets and peptide N-/C-terminal anchor residues[45]. Additionally, H2-D$^b$ bound peptides characteristically form polar interactions within the MHC-I C-pocket, mediated by bi-directional hydrogen bonds between H2-D$^b$ Q$_{97}$ and pN$_5$. The hydrogen bond mediated by H2-D$^b$ Q$_{97}$ leads to its biased presentation of peptides with pN$_5$. Lastly, H2-D$^b$ is defined by a conserved hydrophobic bridge formed by the side chains of W$_{73}$ (α-helix), W$_{147}$, and Y$_{156}$ (α-helix) that runs perpendicular to the binding cleft and imparts an arched solvent-accessible conformation to residues in p6-p8 of H2-D$^b$ bound peptides which is absent from H2-K$^b$ bound peptides[45]. We hypothesised that Hsf2 p.K72N$_{68-76}$, but not WT Hsf2$_{68-76}$, satisfied the pN$_5$ requirement imposed by H2-D$^b$, and that the p6-p8 residues would form a solvent-exposed ridge accessible to the incoming TCR. To validate this notion, we produced the soluble hβ2M/H2-D$^b$/Hsf2 p.K72N$_{68-76}$ (YGFR**N**VVHI) pMHC complex, then crystallised and solved its crystal structure to a resolution of 1.74 Å (Fig. 5b)[46,47]. In the structure, we observed a typical pMHC fold, in which the peptide binding is mainly supported by interactions between the H2-D$^b$ residues at the A, B, D, C and F-pockets and the buried peptide residues pY$_1$-pF$_3$ and pI$_9$, respectively—all of these are conserved between both MT/WT peptides (Supplementary Fig. 7a, b). The A-pocket is occupied by the pY$_1$ side chain, which is solvent facing but remains mostly buried, with the terminal amine engaged in hydrogen bonds with H2-D$^b$ Y$_7$, Y$_{59}$, and Y$_{171}$. The B- and D-pockets are occupied by pG$_2$ and pF$_3$, both of which are also buried within the H2-D$^b$ binding cleft, and pR$_4$, which is solvent exposed (Fig. 5c–e). However, the respective residues exhibit poor compatibility with B- and D-pockets, which are filled with water, possibly weakening binding and increasing thermal flexibility of the pY$_1$-pR$_4$ segment. This notion is supported by the elevated B-factor values associated with the epitope's N-terminus and the adjacent helices in the binary complex structure (Fig. 5c). The hydrophobic F-pocket is occupied by anchor pI$_9$, which is stabilised by van der Waals (VDW) interactions and the H-bonds between its terminal carboxyl group and the H2-D$^b$ N$_{80}$ and K$_{146}$ side chains.

The pN$_5$ side chain is hidden inside the C-pocket and its primary amide is engaged in a hydrogen bond network with the H2-D$^b$ Q$_{97}$ side chain, functioning as a mutant-specific anchor residue (Fig. 4d, e). Such arrangement is typical between H2-D$^b$ and pN$_5$-epitopes and is often observed with non-pN$_5$-peptides (Supplementary Fig. 7c–e), with some exceptions discussed below. We show the superimposition of the H2-D$^b$ structures in complex with Hsf2 NeoAg (PDB 7N9J) or NP-N3D (PDB 4L8C) flu epitope, which share highly similar conformations, despite having only one (pN$_5$)

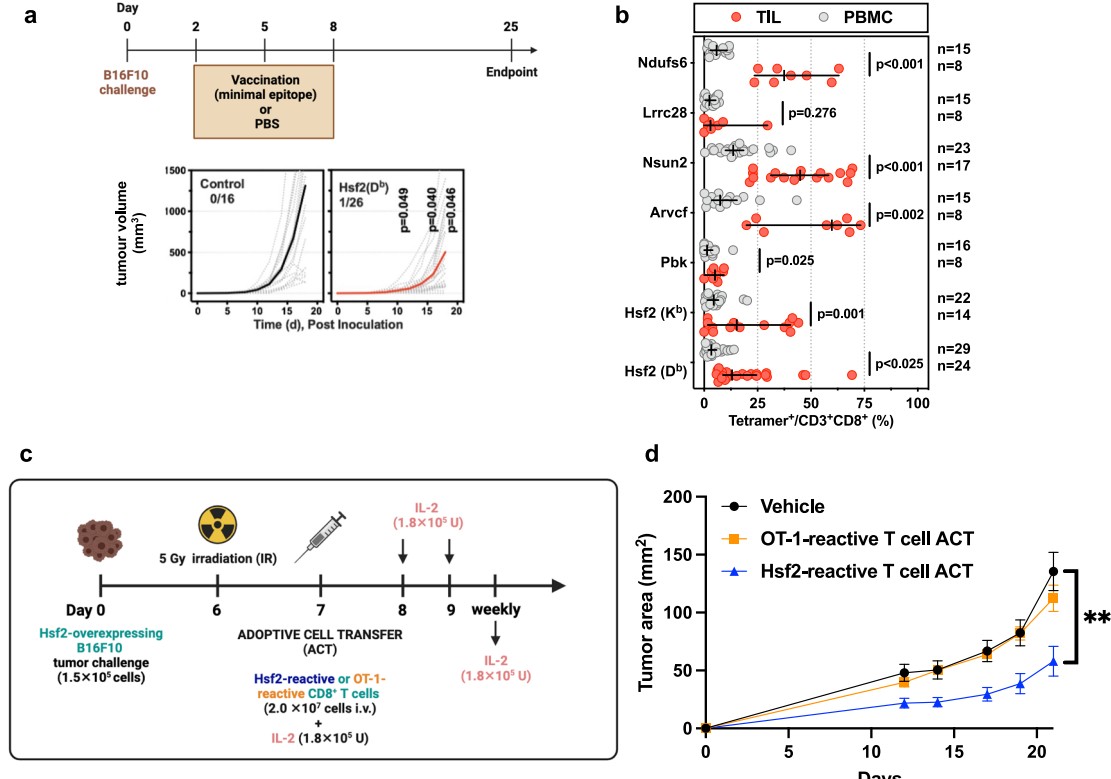

**Fig. 4 | Hsf2 neoantigen-reactive CD8$^+$ T cells elicit anti-tumour activity in vivo.** **a** C57BL/6 mice (*n* = 16 independent biological replicates in control group and 26 independent biological replicates in the vaccinated group) were treated with Hsf2 neoantigen minimal epitope (YGFRNVVHI) vaccine or PBS (mock). Tumours were monitored via calipers. **b** Tetramer staining on CD3$^+$ CD8$^+$ T cells isolated from tumour-infiltrating lymphocytes (TIL) or peripheral blood mononuclear cells (PBMCs) from mice vaccinated with each of the neoantigens listed, as described in 2b. Symbol indicates individual mice, error bars depict the group median, ±95% confidence interval (CI) ± 95% confidence interval (CI), n values (shown on figure) represent independent biological replicates. Statistical analysis consisted of two-sided unpaired *t*-test, followed by Benjamini, Kreiger and Yekutieli two-stage step-

up method, with desired false discovery rate (Q) of 5.00%. Ndufs6: *p* = 0.0000000279, Nsun2: *p* = 0.0000000825, Hsf2: *p* = 0.00053. **c** Schematic (made using Biorender with a full license) describing administration of adoptive cell transfer (ACT) of Hsf2-reactive T cells in vivo. *n* = 15 (vehicle group), 14 (OT-1 group), or 11 (Hsf2 group) C57BL/6 mice (independent biological replicates) collected over 3 independent experiments. **d** tumour growth was measured over time by calipers every 2–3 days and plotted using Graph Pad Prism 7. Error bars depict standard error of the mean. **\*\***p = 0.0036 (vehicle vs. Hsf2-reactive T cell ACT comparison), while for the comparison between OT-1-reactive T cell ACT and Hsf2-reactive T cell ACT, *p* = 0.0122. Two-way ANOVA with Tukey correction. Source data are provided as a Source Data file.

identical residue (Supplementary Fig. 7)[48]. The presence of pN$_5$ usually correlates with increased affinity between peptide and H2-D$^b$. For instance, substitution of pN$_5$ with another residue usually results in a significant affinity drop between such peptides and H2-D$^b$ which correlates with our data presented in Fig. 5a[49].

The C-terminal peptide side chains−pR$_4$ and pV$_6$-pV$_7$-pH$_8$−are solvent-exposed and do not directly contribute to MHC-I binding. Residues pV$_6$ and pV$_7$ form an arch over the W$_{73}$-W$_{147}$-Y$_{156}$ bridge that projects towards A$_{152}$ of the α$_2$-helix, with pH$_8$ projecting back towards the same helix. This type of arrangement is common for H2-D$^b$ and mostly observed independent of the peptide amino acid sequence (Supplementary Fig. 7c−e) with a possible exception of the peptides with nonpolar or aromatic side chains at positions p$_5$ and p$_6$[50].

Analysis of crystallographic B-factors revealed asymmetry in the distribution of peptide all-atom flexibility with N-terminal residues proximal to the pN$_5$ anchor exhibiting increased average B-factor values relative to distal C-terminal residues, pY$_1$-pR$_4$ [39.05 ± 4.97] versus pN$_5$ − pI$_9$ [27.64 ± 1.56] (Fig. 5c). Overall, this arrangement suggests H2-D$^b$/Hsf2 p.K72N$_{68-76}$ is stabilised primarily at the C-terminus, via the contribution from conserved pI$_9$ anchor as well as the pN$_5$ anchor. Additionally, the increased thermal rigidity of the C-terminal region and elevated solvent exposure altogether suggest that the pY$_1$-pR$_4$ epitope segment may be preferentially targeted by TCRs[45].

In summary, the H2-D$^b$/Hsf2 p.K72N$_{68-76}$ crystal structure revealed a typical H2-D$^b$ peptide binding mode mediated by hydrophobic interactions between residues lining the H2-D$^b$ A-B-D and F-pockets and peptide N-/C-terminal anchor residues[45]. Additionally, the H2-D$^b$ bound epitope with pN$_5$ supports bi-directional hydrogen bonds between the H2-D$^b$ Q$_{97}$ and peptide anchor pN$_5$ side chains. Thus, the K72N mutation in *Hsf2* gene has produced a prototypical group II neoAg with high affinity for H2-D$^b$. In contrast, the WT Hsf2 peptide had low affinity toward H2-D$^b$.

### TCR-47BE7 preference for Hsf2 p.K72N$_{68-76}$

Having identified a plausible physical mechanism for enhanced binding of the neoAg Hsf2 p.K72N$_{68-76}$ to H2-D$^b$, we sought to better understand the preference of the 47BE7 TCR for the MT peptide K72N$_{68-76}$ observed in earlier functional studies (Fig. 2c, d). Selectivity for the MT peptide is an important characteristic of neoAg-reactive TCRs that could provide effective tumour growth control with minimal off-target effects relative to that observed upon treatment with TCRs responsive to tumour-associated antigens[35]. In cell-based cytokine production assays, we found TCR 47BE7 to be approximately 1.55×10$^6$ fold more sensitive to the MT peptide (EC$_{50}$, 5.6pM) relative to the WT peptide (EC$_{50}$, 8.7 μM) (Fig. 2d). This difference was several orders of magnitude larger than the 175-fold

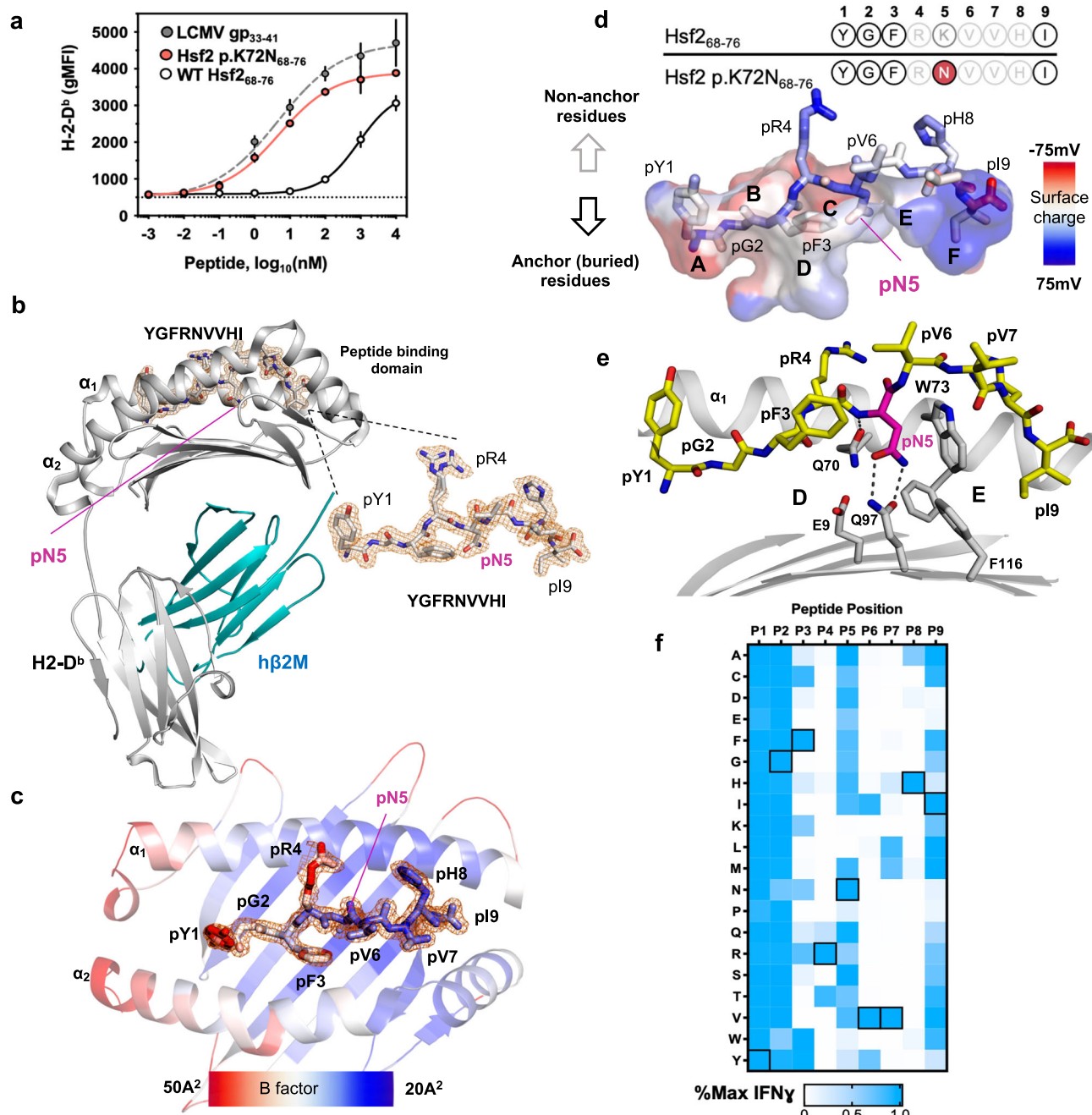

**Fig. 5 | Structure of the pMHC binary complex.** In selected figures, one alternate conformation for pR4 was excluded for clarity. **a** RMA-S cells were plated at 25 °C for 18 h, then co-incubated with the indicated peptides for 30 min at 30 °C, followed by 3 h at 37 °C. Surface H2-Db geometric mean fluorescence intensity (gMFI) was measured by flow cytometry and plotted for peptide concentrations tested. Symbol indicates the mean of biologic replicates ($n = 3$), ±95% confidence interval (CI). **b** Crystal structure of H2-Db/Hsf2 p.K72N68-76. H2-Db is coloured in grey, human β2M (hβ2M) is shown in blue. pN5 refers to the asparagine (N) located at p5. Peptide stick model: oxygen coloured red, nitrogen is blue and carbon is grey. SigmaA-weighted Fo-Fc electron density map ($\delta = 4.0$, radius = 1.5 Å) is shown around peptide. **c** The peptide binding domain of H2-Db (cartoon) and Hsf2 p.K72N68-76 are coloured according to B-factor values. Peptide residues shown as sticks. The SigmaA-weighted 2Fo-Fc map ($\sigma = 1.0$, radius = 1.5 Å) is superposed onto peptide residues only. **d** Peptide-binding cavity of H2-Db is shown as a surface area coloured according to the surface charge. Hsf2 p.K72N68-76 residues are shown as sticks, with atoms coloured according to charge. The approximate location of each binding pocket is marked by a letter from A (N-terminal pocket, residue P1) to F (C-terminal pocket, residue P9). Top – amino acid sequences for mutant (MT) and wild type (WT) Hsf2 peptide, anchor and buried residues (SASA < 20%) are depicted in bold. **e** Arrangement of residues in the C-pocket of the binary complex. Cartoon and stick model. Hsf2 p.K72N68-76 carbon atoms are coloured in yellow, H2-Db carbons are coloured in grey. Oxygen atoms are red, nitrogen, blue. H-bonds are dotted lines. **f** Peptide scan. CD8+TCR-47BE7+ T cells were incubated with 1 μM of each YGFRNVVHI peptide variant from a positional scanning library, along with αCD28, for 6 h and IFNγ production measured by intracellular flow cytometry staining (gating strategy shown in Supplementary Fig. 8). Colour indicates mean of biologic replicates ($n = 2$). Boxed squares indicate native amino acid at indicated position within mutant peptide YGFRNVVHI. Source data are provided as a Source Data file.

difference observed in the RMA-S MHC-I stabilisation assay (Fig. 5a). Moreover, in response to saturating peptide concentrations, we observed significantly greater cytokine production on a per-cell basis when exposed to MT versus the WT peptide (Fig. 2c), suggesting that the WT peptide is a weak agonist. Altogether, these data indicated that the MT and WT Hsf2 peptides are discriminated by TCR 47BE7 as well as by H2-D$^b$.

To further characterise the biochemical basis for antigen discrimination by TCR 47BE7 we generated a positional scanning peptide library wherein each position within Hsf2 p.K72N$_{68-76}$ was replaced with each of the remaining 19 protein-coding amino acids, and then assessed cytokine production by 47BE7-expressing tgTCR CD8$^+$ T cells (Fig. 5f, Supplementary Fig. 8). The majority of side-chain substitutions of the non-core anchor residues pY$_1$-pG$_2$ and pI$_9$ were generally tolerated. A limited set of conservative, primarily aromatic substitutions was tolerated at p3, rendering these residues non-essential for TCR 47BE7 binding. Conversely, the vast majority of side-chain substitutions within the core epitope pR$_4$-[X]-pV$_6$-pV$_7$-pH$_8$ abolished TCR recognition and therefore were considered essential for TCR 47BE7 binding. Exceptions included biochemically conservative substitutions for the polar-basic residue pR$_4$ (replaced with Q, T) and non-polar residues pV$_6$ (replaced with I, Y) and pV$_7$ (replaced with L, M). Notably, TCR 47BE7 tolerated multiple side-chain substitutions at p5. This observation could be explained by both the lack of significant interfacial contacts between TCR 47BE7 and the side chain of pN$_5$, as well as that other residues at p5 allow the same or similar conformation of the core neoepitope (Supplementary Fig. 7a). Consistent with this, comparison of available H2-D$^b$ structures demonstrates that alternative p5 residues, including glycine, alanine, aspartate, histidine and methionine, adopt anchor conformations similar to that of asparagine (Supplementary Fig. 7e). In our studies, the exceptions from the above rule were the two substitutions at p5, lysine and leucine, respectively. The corresponding mutant peptides were unable to produce detectable activation of the TCR 47BE7-expressing T cells in the given conditions (Fig. 4f). Because strong peptide-MHC interaction is necessary for TCR binding and T cell activation[51–53], we performed in silico binding affinity analysis of all position 5 substituted peptides using NetMHCPan4.1. We observed a weak but significant direct correlation between predicted affinity to MHC-I and the TCR 47BE7 activation (Supplementary Fig. 7f). Notably, the WT Hsf2$_{68-66}$ (YGFRKVVHI) was an outlier with no T cell activation detected, suggesting that peptide-MHC binding affinity alone was not sufficient to explain the selectivity of TCR-47BE7.

Using the peptide scan data presented in Fig. 4f, we created a ProSite search pattern - x-x-[CRFKWY]-R-{KL}-[IV]-[VML]-[HA]-[ILMVCAFWT][54] and utilised this pattern to identify sequences in mouse proteome potentially cross-reactive with TCR 47BE7. The initial search produced a total of 107 hits (Supplementary Data 1). Based on affinity estimates by NetMHCPan4.1, only one peptide satisfied the epitope selection criteria (NVFRNILHV, Uniprot ID Q9D3N2, with an estimated K$_D$ value below 1 μM). However, comparison with the mouse immunopeptidome ruled out this epitope, as it was not detected in complex with H2-D$^b$ in published mouse tissue analysis databases[55].

In summary, the peptide scan analysis combined with the crystal structure of the binary complex suggest that the core pattern R$_4$-X-V$_6$-V$_7$-H$_8$ of the Hsf2 p.K72N$_{68-76}$ epitope is selectively recognised by TCR 47BE7. The lack of significant similarity between the R$_4$-X-V$_6$-V$_7$-H$_8$ pattern and the mouse proteome and immunopeptidome renders the possibility of significant off-targeting by TCR 47BE7 unlikely. These data support the notion that Hsf2 p.K72N$_{68-76}$ could behave as a "non-self" epitope, triggering a strong immune response not curtailed by immune tolerance. To understand the molecular mechanism of immunogenicity, we then determined the crystal structure of the ternary complex between TCR 47BE7 and H2-D$^b$/Hsf2 p.K72N$_{68-76}$.

## Crystal structure of ternary complex

Given our structural data detailing the enhanced binding of the neoAg Hsf2 p.K72N$_{68-76}$ to H2-D$^b$, we sought to understand the mechanism for its recognition by cognate TCR 47BE7.

We hypothesised that TCR recognition was mediated by the interactions between the TCR and mutant peptide-specific structural features of the pN$_5$ anchor and the solvent-exposed p6-p8 ridge. For these studies, we produced recombinant soluble TCR 47BE7 as previously described[56]. Correct folding and preserved substrate recognition in solution were determined by measuring binding kinetics between TCR 47BE7 and H2-D$^b$/Hsf2 p.K72N$_{68-76}$ by biolayer interferometry (BLI). The 47BE7 TCR binds immobilised H2-D$^b$/Hsf2p.K72N$_{68-76}$ with a high affinity typical for non-self-reactive TCR (K$_D$ 2.7 ± 0.3 μM) (Fig. 6a)[57]. The on and off-rates for complex binding were too fast to be calculated with precision. To determine the structural basis for epitope recognition and T cell activation, we crystallised and solved the TCR-47BE7/H2-D$^b$/Hsf2 p.K72N$_{68-76}$ ternary complex structure to a resolution of 2.5 Å (Fig. 6b, Supplementary Table 2). Our data show that bound Hsf2 p.K72N$_{68-76}$ peptide was well defined on the electron density map, adopting nearly the same conformations in the binary and ternary complexes, respectively (0.6 Å Cα RMSD, Fig. 5b and Supplementary Fig. 9a). The well-defined electron density at the TCR:pMHC interface allowed for unequivocal placement of all critical amino acid side chains at the interface (Supplementary Fig. 9b). 47BE7 exhibited conventional oblique docking geometry (13.13° incident angle, 57.49° docking angle)[58]. Notably, the TCR centroid was biased towards the peptide C-terminus, spanning the p6-p8 core segment identified in the binary structure and predicted to contribute to TCR binding. Binding between the TCR and the Hsf2 p.K72N$_{68-76}$ peptide was mediated by the C-terminal half of epitope comprised of pR$_4$-[X]-pV$_6$-pV$_7$-pH$_8$. In agreement, a comparison of solvent-exposed surface area (SASA) of the bound and unbound pMHC complex shows significant SASA reduction for pR$_4$ and pV$_6$-pH$_8$, which comprises the peptide contribution to the core epitope buried at the TCR-pMHC interface (Fig. 6c). For the remaining peptide residues - including the mutated residue pN$_5$ - SASA did not change significantly on complexation, indicating that these side chains remain buried or otherwise do not contribute to ternary complex formation. The buried residue H2-D$^b$ Y$_{156}$ projected into the E pocket in the pMHC structure was observed to rotate towards the D pocket in the bound structure, and the space previously occupied by Y$_{156}$ is instead filled with glycerol (Supplementary Fig. 9d, Fig. 6a). The presence of a glycerol contaminant in this position did not significantly alter conformations of other amino acid side chains, including the p6-p8 arched peptide conformation, and had no appreciable impact on the TCR-pMHC interface, but only reflected the flexibility of H2-D$^b$ Y$_{156}$. The mobility of Y$_{156}$ observed in these structures could expand the size of the E pocket (Supplementary Fig. 9a–c), so it may accommodate the aromatic side chains of epitope residues at p5 or p6 including phenylalanine or even tryptophan[50] (Supplementary Fig. 9d, e, Supplementary Fig. 12).

The total area buried upon complexation between pMHC and TCR (1506 Å$^2$) was almost evenly divided between the two components (Fig. 6d, e). Interfacial contacts between 47BE7 and H2-D$^b$/Hsf2 p.K72N$_{68-76}$ were mediated by complementarity determining region (CDR) loops CDR1α, CDR2α, CDR3α, CDR2β, and CDR3β, with little contribution from CDR1β (Fig. 6d, g, f). Binding of the TCR to H2-D$^b$ was mediated by the solvent-exposed residues Q$_{72}$, R$_{75}$, R$_{79}$ (H2-D$^b$ α$_1$ helix), as well as E$_{18}$ (H2-D$^b$ loop A), which undergo re-organisation upon complexation (Fig. 5g and Supplementary Fig. 9e–g), placing E$_{18}$ within the hydrogen bond distance of R$_{75}$ and R$_{79}$ and positioning R$_{75}$ to form hydrogen bonds with the hydroxyl and carboxyl groups of CDR2β S$_{51}$ and the Y$_{52}$ backbone carboxyl, respectively (Supplementary Fig. 9f). Additional interfacial TCR-pMHC polar interactions include hydrogen bonds between CDR2β M$_{56}$ and Q$_{72}$, CDR3β E$_{97}$ and N$_{80}$, as well as salt bridges between CDR3β E$_{97}$ and K$_{146}$ (H2-D$^b$ α2

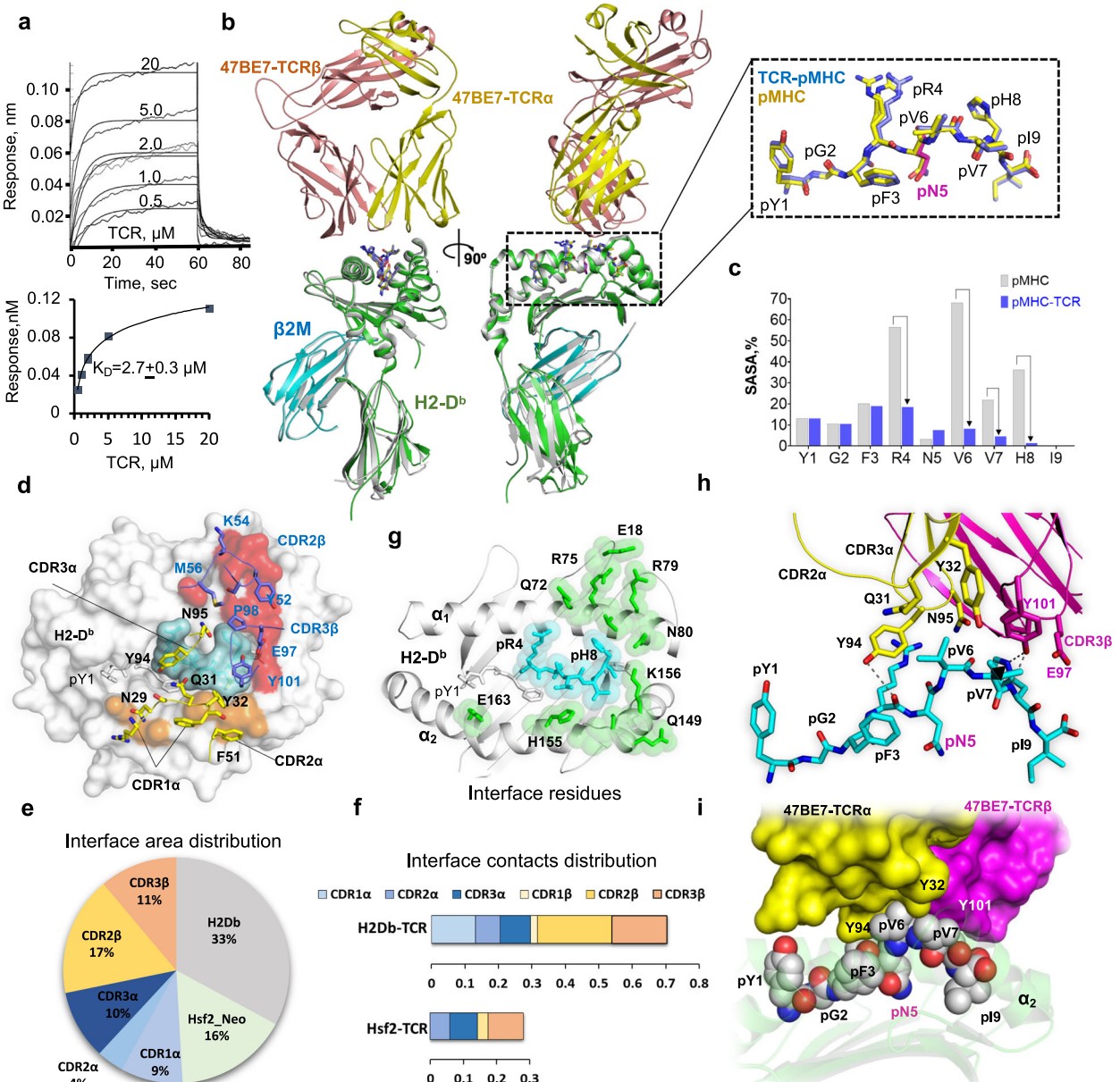

**Fig. 6 | Structure of the pMHC-TCR ternary complex. a** Kinetics of soluble TCR-47BE7 binding to immobilised H2-D$^b$/Hsf2 p.K72N as determined by biolayer interferometry. The dissociation constant ($K_D$) was determined by curve fitting in Octet® 9.1 System Data Analysis software. **b** Structural superimposition of TCR-47BE7/H2-D$^b$/Hsf2 p.K72N$_{68-76}$ (7NA5) and H2-D$^b$/Hsf2 p.K72N$_{68-76}$ (7N9J) structures. (Inset) Superposition of the corresponding Hsf2 p.K72N$_{68-76}$ peptides. **c** Residue-specific solvent-accessible surface area (SASA) in unbound/binary (grey) and bound/ternary (blue) structures were calculated using NACCES[89]. **d** TCR footprint on pMHC (inter-atomic distance cutoff 4 Å). En face view of H2-D$^b$/Hsf2 p.K72N$_{68-76}$ with superimposed TCR-47BE7 CDR loops with residues as sticks. H2-D$^b$ interface is coloured according to observed CDR1α-CDR3α (orange), CDR1β-CDR3β

(red), or no TCR contact (white). Peptide interface is coloured blue. **e** TCR-pMHC interface distribution between components. **f** Distribution of inter-atomic interface contacts between TCR and pMHC, distance cutoff 8 Å. Computed by NACCESS, as described in Methods. **g** Interface H2-D$^b$ (green) and peptide (blue) residues, stick models. Inter-atom distance cutoff 4 Å. **h** The interface between bound neoepitope and TCR-47BE7 in the pMHC-TCR structure in cartoon model. The polar bonds are depicted as dotted lines. Only the TCR residues that are in direct contact (<4 Å distance) with epitope atoms are shown as sticks. **i** Interface between Hsf2 p.K72N$_{68-76}$ (atoms presented as VDW spheres) and TCR-47BE7 (the protein surface was coloured according to the TCR chains). One alternate conformation for pR$_4$ was excluded for clarity. Source data are provided as a Source Data file.

helix). Polar interactions between TCRα and H2-D$^b$ were limited to a single hydrogen bond between CDR1α Y$_{32}$ and S$_{150}$ (H2-D$^b$ α2 helix) (Supplementary Fig. 9g).

The polar interactions between the peptide and TCR involved the CDR3α Y$_{94}$ hydroxyl and the pR$_4$ backbone carboxyl; the CDR3β Y$_{101}$ hydroxyl and the pH$_8$ side chain imidazole and backbone amide; as well as a salt bridge between CDR3β E$_{97}$ and pH$_8$. Other electrostatic interactions were observed in the form of water bridging between CDR2β D$_{57}$ as well as CDR3α N$_{95}$ and pR$_4$. Finally, non-

polar van der Waals (VDW) contacts were between CDR1α Q$_{31}$ and Y$_{32}$ and pV$_6$ and, as well as between CDR3α Y$_{32}$, CDR3β Y$_{101}$ and pVal$_7$ (Fig. 5h, i). Collectively, our data illustrate that the TCR 47BE7-epitope interaction is biased towards the peptide C-terminus, in a region of pre-existing structural rigidity within H2-D$^b$/Hsf2 p.K72N$_{68-76}$ binary complex. While the mutant pN$_5$ residue lies within the core pR$_4$-[X]-pV$_6$-pV$_7$-pH$_8$ epitope, it makes no significant side-chain contacts with TCR 47BE7, confirming its minimal direct impact on TCR binding.

The structure of TCR 47BE7:H2-D$^b$/Hsf2p.K72N shares common features with published TCR-H2-D$^b$ complexes found in the PDB and TCR3D databases ($n = 16$)[58]. Despite structural similarities, each TCR adopts a distinct orientation with respect to cognate binary complex, with interface area or the TCR docking angle values largely varying, with the most similarity observed between the 47BE7 structure (7NA5) and (7N4K) (Supplementary Fig. 10a, b). The conformation of each binary complex was minimally affected by the nature of the bound epitope, where each peptide formed a typical p6-p8 C terminal arch (Supplementary Fig. 10c). The larger docking angle value inversely correlated with smaller interface area (Supplementary Fig. 10b), but the latter usually does not correlate with TCR avidity or affinity[59]. Instead, the smaller buried interface in the 47BE7 structure supports a significant contribution from the bound epitope, potentially increasing TCR selectivity.

The complex between the WT peptide and H2-D$^b$ was unstable and not available for further analysis. Consequently, the molecular basis for discrimination between the WT and mutant Hsf2$_{68-76}$ peptides remains unclear, although a few plausible scenarios exist, corroborated by the available biochemical data and molecular modelling. Our study indicated that the WT peptide has low affinity towards H2-D$^b$ (Fig. 5a). Other pK$_5$ peptides follow a similar trend. For instance, the affinity for pN$_5$ peptides was significantly higher as compared with the non-pN$_5$ peptides, whereas all the pK$_5$ peptides displayed low affinity with IC$_{50}$ > 1 µM (Supplementary Fig. 11a). It is noteworthy, that among 2276 H2-D$^b$-specific 9-amino acid (AA) peptides representing mouse immunopeptidome, only 3 (-0.1%) had lysine residue at p$_5$. Further, the WT Hsf2$_{68-76}$ epitope was not detected in any mouse tissue[55].

To determine the structural basis for discrimination between pK$_5$ and pN$_5$ peptides, we performed in silico docking of WT Hsf2$_{68-76}$ using PepFlexDoc and the crystal structures we solved as templates. In total we have utilised the following models: (N5) H2-D$^b$ structure with the MT epitope as a benchmark, (K5a) H2-D$^b$ structure in which pN$_5$ was replaced with lysine in anchor orientation, (K5na) H2-D$^b$ structure in which pN$_5$ was replaced with lysine in a non-anchor orientation, and (K5a_Y156a) H2-D$^b$ structure in which pN$_5$ was replaced with lysine in anchor orientation and with Y$_{156}$ adopting the alternate conformation observed in the ternary complex. Each model was reduced to the H2-D$^b$ domain 1 (AA residues 1–180) and peptide ligand. The top 10 docking scores from each of these scenarios are plotted, with the N5 scenario scoring the highest (Supplementary Fig. 11b). Other scenarios produced more widespread and overlapping solutions, with K5a-Y156a being the least plausible. Comparison of the top 10 solutions from scenarios K5a and K5na represent the two alternate conformations for the H2-D$^b$/Hsf2$_{68-76}$ complex. The pK$_5$ non-anchor orientation (K5na) resulted in docking solutions with WT peptide conformations distinct from that of MT epitope (Supplementary Fig. 11c, d), whereas the H2-D$^b$ residues retained their original conformations. By contrast, the pK$_5$ anchor orientation (K5a) resulted in the WT peptide conformations similar to that of MT epitope found in the crystal structure. However, to accommodate the pK$_5$ side chain inside the binding pocket and to avoid inter-atomic clashes, the bulky aromatic side chain of H2-D$^b$ W73 required drastic conformational change (Supplementary Fig. 11e, f). Based on the docking scores, both scenarios remain possible, but probable modelling bias must be accounted for. The body of evidence indirectly favours the non-anchor pK$_5$ conformation, as rotation of a solvent-buried tryptophan side chain toward exposed orientation seems unlikely and has not been previously been observed in H2-D$^b$. Moreover, HLA or H2 molecule promiscuity is largely the result of peptide structural plasticity combined with limited specificity, defined as peptide binding motifs[59]. In any case, the modelling and docking experiments clearly indicate that the H2-D$^b$ C pocket is unable to accommodate the extended and basic pK$_5$ side chain without structural re-arrangements eliminating steric clashes between peptide and the H2-D$^b$ residues lining the peptide binding site. Thus, binding of WT Hsf2$_{68-76}$ may yield either a distinct peptide conformation or distortion of the H2-D$^b$ binding pocket structure. In such scenarios, the interface between H2-D$^b$/WT Hsf2$_{68-76}$ and TCR 47BE7 would be severely compromised due to clashes between either exposed pK$_5$ (model K5na) or between the exposed H2-D$^b$ W73 side chain (model K5a) from one side and TCR residues from another.

Collectively, we determined the molecular mechanism of immunogenicity for the H2-D$^b$-restricted B16F10 melanoma neoAg Hsf2 p.K72N$_{68-76}$. This neoepitope is discriminated from the WT peptide at the MHC level due to a K72N amino acid substitution at the anchor position p5 and can be classified as a group II neoAg[27]. At the same time, it is discriminated at the TCR level, as the H2-D$^b$/ Hsf2 p.K72N$_{68-76}$ complex adopts a unique conformation specifically recognised by TCR 47BE7, whereas the WT peptide is not efficiently recognised. In addition, the earlier published data show an apparent lack of epitopes cross-reactive to Hsf2 p.K72N$_{68-76}$ across the mouse immunopeptidome. Based on all of the above, we conclude that the neoAg Hsf2 p.K72N$_{68-76}$ is a unique "non-self", prototypical neoantigen in B16F10 melanoma.

## Discussion

Neoantigens (neoAg) have garnered significant interest as therapeutic targets due to their potential to exhibit enhanced therapeutic efficacy and safety. While studies have described features of immunogenic neoantigens based on post hoc computational analysis[60], detailed molecular and structural studies of neoAg and corresponding TCR remain limited. Our study aimed to address this by investigating fundamental assumptions regarding the molecular and structural determinants of neoantigen immunogenicity. We did so in the B16F10, a pre-clinical model with a well-characterized response to existing immunotherapies and, importantly, with other known antigens available for future comparative studies. Previously, many of the B16F10 antigens studied were either tumour-associated antigens (typically melanocyte differentiation antigens such as gp100 or Trp2)[40–42] or OVA$_{257-264}$, a foreign antigen engineered to be artificially expressed by B16F10[32]. Others have investigated neoantigens in B16F10, uncovering immunogenic epitopes (distinct from those discovered in our study) that elicit tumour growth reduction upon peptide vaccination[61]; however, extensive analyses of neoAg-reactive TCRs in B16F10 and their structure-function relationships were not performed previously.

We first identified and characterised several TCR-antigen pairs from the widely-utilised B16F10 melanoma model. We then completed in-depth biochemical and structural studies of the prototype anchor residue-modified neoantigen H2-D$^b$/Hsf2 p.K72N and the corresponding monoclonal TCR 47BE7. While H2-D$^b$/Hsf2 p.K72N has been identified in a previous analysis of B16F10[60], we were able to demonstrate its immunogenicity, identify an Hsf2-reactive TCR, and conduct its comprehensive structure-activity relationship (SAR) characterization. We selected H2-D$^b$/Hsf2 p.K72N$_{68-76}$ and 47BE7 for characterization due to demonstrable in vitro and in vivo activity in a challenging tumour model. Importantly, while vaccination with Hsf2 p.K72N$_{68-76}$ peptide suppressed tumour growth, ACT shows significant anti-tumour effect only with Hsf2 p.K72N$_{68-76}$ overexpressed by B16F10 cells. This divergence indicates that the intrinsic level of neoAg expression is not sufficient for tumour suppression using ACT alone. This observation is in accordance with published data demonstrating that antigen-transduced (OVA, Pmel-1) B16F10 is highly resistant to ACT and often requires combinatorial strategies (i.e., checkpoint blockade) and/or overexpression or peptide/TCR modifications to impart efficacy[33,34,62]. For the purposes of this study, vaccination and ACT serve as proof-of-concept that various immunotherapeutic modalities involving Hsf2 p.K72N can yield tumour growth control in vivo. Given that 47BE7 exhibited high functional avidity and limited cross-reactivity in our in vitro studies, we hypothesised this model

could provide insight into mechanisms of B16F10 resistance to immunotherapy.

We found that the lysine to asparagine (p. K72N) anchor residue substitution results in a 175-fold improvement in the surface presentation of H2-D$^b$/Hsf2 p.K72N$_{68-76}$ compared to that of the wild-type Hsf2$_{68-76}$. The crystal structure of H2-D$^b$/Hsf2 p.K72N$_{68-76}$ demonstrates that the mutated pN$_5$ residue is directly responsible for this effect, due to stabilizing polar interactions between the Hsf2 p.K72N epitope and H2-D$^b$ Q$_{97}$. This anchor residue mutation also induced the formation of a rigid, solvent-exposed, hydrophobic arch at the carboxy-terminal pV$_6$-H$_8$ segment of the Hsf2 p.K72N$_{68-76}$, which was essential for binding to TCR 47BE7. The structural stability of neoAg pMHC has been repeatedly associated with immunogenicity, often attributed to increased pMHC surface abundance due to slow peptide dissociation kinetics[45]. Our data reaffirm and extend this finding, showing that stability measures may indirectly capture fine structural features within an epitope that contribute to immunogenicity through TCR contacts, exemplified by the rigid hydrophobic pV$_6$-pH$_8$ arch.

Recently, several groups have published structural studies of human neoAg-reactive TCR[27–30,63,64]. The structural data we present support and expand on these earlier findings in several noteworthy ways. First, we observed high-level commonalities between TCR 47BE7/Hsf2 p.K72N and TCR9a/TCR10, as well as TCR4, which bind to the group II (anchor-residue modified) neoAg HLA-C*08:02/Kras p.G12D[27] These TCRs employ a similar binding mode characterized by multiple intermolecular contacts distributed across the TCR:pMHC interface. As in TCR 47BE7, experimental modification of contact residues within the core TCR epitope eliminates TCR reactivity, suggesting that the totality of the interface is necessary for TCR binding[27]. This binding mode often contrasts with that employed by neoAg-reactive TCR that bind to group I (solvent-exposed residue modified) neoAg such as TCR12-6/TCR38-10, which bind to the neoAg HLA-A*02:01/TP53 p.R175H[29]. In this latter circumstance, the observed TCR contacts are biased towards the solvent-exposed mutant residue and avoid contacts with the remaining peptide surface. While we did not perform structural characterization of our 29BF8 TCR (recognizing H2-K$^b$ Lrc28 p.K329T) or our 44CH2 TCR (recognizing H2D$^b$ Pbk p.V145D), we would hypothesise that they employ a similar binding mechanism to Group I neoAgs. Further elucidation of the Group I and Group II neoAgs and their respective TCRs may further guide identification of effective neoAg-specific TCRs in the future.

We acknowledge several limitations of our study, primarily being restricted to a murine model system, which limits direct translation to the human context. We also focused on a single neoantigen-TCR pair, in the context of a single MHC-I allele, and while we note that ~40% of H2-D and H2-L alleles share similar structural features, this possibly limits generalizability outside of this context. However, we have created a model system of neoAgs in a commonly used preclinical model; this may enable study of low- vs high-affinity neoAgs and cross-reactive vs mutant peptide-specific TCRs alike, thus enabling us to create an atlas of neoAg pMHC-TCR interactions and their relationship to immunotherapy efficacy. More specifically, our finding that our highest affinity TCR, 47BE7, but not other neoAg-reactive TCRs, can recognise tumours and control tumour growth confirms the importance of high affinity TCRs in enabling recognition of neoAgs, which despite their high theoretical immunogenicity, are often expressed at a lower level than TAAs in tumours[65]. Future studies should perhaps consider selecting high avidity TCRs that enable potent anti-tumour immune responses, provided that these TCRs are not cross-reactive with healthy tissue or cause signalling fatigue, as has been previously observed[66–68]. Further, we have identified a neoantigenic structural motif in the context of H2-D$^b$, which supports strong pMHC-TCR interactions that mediate robust complex stability and is the basis of 47BE7's high affinity for its cognate pMHC. While this may seem unique to our model, as described above, we observe commonalities with human TCRs previously studied and provide evidence that TCRs exhibiting similar features to ours will be good candidates for vaccination with pertinent neoAg peptides and perhaps for adoptive cellular therapies. Further, using the SAR approach, we fully validated the group II neoAg Hsf2 p.K72N$_{68-76}$ and determined the mechanism of its immunogenicity at the molecular level. Altogether, we have identified neoAgs and neoAg-reactive T cells in B16F10 and have provided functional and structural data on a high affinity TCR, 47BE7, that confers tumour protection and could inform the study and identification of human neoAg-reactive TCR biology in the future.

## Methods

All research complies with ethical regulations. Mouse procedures and monitoring protocols were approved by the Icahn School of Medicine at Mount Sinai Institutional Animal Care and Use Committee (IACUC) protocol: 15-2171, approval: IACUC-2016-0028.

### Cell culture

B16F10 (cat# CRL-6475) and RMA-S (cat# TIB-39) cells were purchased from American Type Tissue Culture (ATCC). Platinum-E (cat# RV-101) cells used for retroviral packaging were purchased from Cell Biolabs. Upon arrival, cells lines were tested regularly for mycoplasma (Lonza, cat#: LT07-318), and rodent pathogens by IMPACT (IDEXX), and reference cell banks were generated. All cell lines were maintained Dulbecco's Modified Eagle's Medium (DMEM) with GlutaMAX™, HEPES 20 mM, penicillin-streptomycin and fetal bovine serum (FBS) 10%v/v at 37 °C in a 5% CO$_2$ humidified atmosphere.

### Whole exome sequencing

B16F10 (tumour) cells were expanded in culture to 75% confluence. Total splenocytes (germline) were isolated from a male C57BL/6 colony founder. Genomic DNA was isolated using DNeasy Blood & Tissue kit (Qiagen, cat#: 6950). Whole exome sequencing (WES) libraries were prepared using SureSelectXT Mouse All Exon kits (Agilent, cat# G7550A). Paired-end 100 bp sequencing was performed using HiSeq2500 reagent kit v3 (Illumina, CA) targeted sequencing depths of 300x and 150x for tumour and germline samples, respectively. Sequencing reads were mapped to GRCm38.p6/mm10 using BWA-MEM[69]. Duplicate read marking and base quality score recalibration were performed using GATK/Picard[70]. Somatic variant calling was performed for target regions using MuTect and Strelka with default filters[71].

### Isolation of tumour mRNA and RNA sequencing

B16F10 cell tumour cells ($1 \times 10^6$) were inoculated into the dermis of subject animals. Seven days post-inoculation the tumours were resected, and total RNA was isolated using RNeasy kits (Qiagen, cat#: 74104). Messenger RNA sequencing library generation was performed using Ribo-zero magnetic gold and TruSeq RNA Sample preparation kits (Illumina, CA). Paired-end, 100 bp, sequencing was performed using a HiSeq 2500 reagent v3 kit, with a targeted sequencing depth of $1 \times 10^8$ reads/library. Sequencing reads were mapped to GRCm38.p6/mm10 using HiSa.

### Identification of mutation-derived tumour neoantigens (neoAg)

Mutation-derived tumour neoantigens were identified using our established pipeline[36]. Briefly, somatic variants are identified by WES. Variant expression is quantified by local assembly and allele-specific quantification of mutated and reference transcripts. Variant transcripts are translated in silico. The peptide-MHC-I binding prediction tool NetMHCpan (v.4.1) was then used to identify candidate neoAg for further study. Code is available at https://github.com/openvax/vaxrank.

## Peptide synthesis

Experimental peptides were individually custom synthesised via the solid-phase method and validated by GenScript (Piscataway, NJ), with standard removal of trifluoracetic acid and replacement with hydrochloride, purified to >98% by HPLC, and lyophilised for storage. Peptides were reconstituted in DMSO at 10 μM and frozen at −80 °C until use.

## Immunization

Peptide-based vaccines comprising peptide antigen and TLR7/8a adjuvant co-delivered in self-assembling particles (referred to as "SNAPvax™") were produced in accordance with standard protocol[38,39]. Briefly, peptide antigens were synthesised and were linked to imidazoquinoline-based TLR7/8a (Barinthus Therapeutics, USA) using an azide-alkyne cycloaddition click chemistry reaction. Vaccines were reconstituted in sterile phosphate-buffered saline to a final concentration of 40 μM, and 50 μL was injected subcutaneously to bilateral footpads.

## Mice

C57BL/6 J (C57BL/6) and C57BL/6-Tg (Nr4a1-EGFP/cre) 820Khog/J (Nr4a1-eGFP) were purchased from Jackson Laboratory (Bar Harbour, ME). Mice were housed in a specific pathogen-free (SPF) containment facility (12-h light/12-h dark cycle, 21–22 °C, 39–50% humidity) located at the Icahn School of Medicine at Mount Sinai. Procedures and monitoring protocols were approved by the Icahn School of Medicine at Mount Sinai Institutional Animal Care and Use Committee (IACUC) protocol: 15-2171, approval: IACUC-2016-0028.

8–12-week-old animals balanced and completely randomised with respect to age and sex, were used for all immunization, adoptive cell transfer, and tumour allograft experiments. Subjects were evaluated every 48 h for the full duration of all experiments. Euthanasia was performed by carbon dioxide asphyxiation followed by cervical dislocation.

Peripheral blood was obtained by submandibular vein puncture. Approximately, 250 μL of was collected into sterile heparinised tubes. Red blood cells were removed with ammonium-chloride-potassium (ACK) osmotic lysis solution; 2:1 v/v for 5 min, followed by centrifugation at 500 × g, 5 min. The resulting peripheral blood mononuclear cells (PBMC) were washed twice with MACS buffer (PBS, BSA 2%v/v, EDTA 2 mM), and then stored in MACS buffer at 4 °C until use.

Total splenocytes were obtained by splenectomy. Tissue samples were macerated over 70 μm pore-size nylon filters. Red blood cells were removed by treating the samples with ACK lysis solution; 2:1 v/v for 5 min, followed by centrifugation at 500 × g, 5 min. Total splenocytes were washed twice with MACS buffer (PBS, BSA 2%v/v, EDTA 2 mM) and maintained at 4 °C until use.

## Tumour inoculation

Orthotopic injections of B16F10 melanoma cells were performed[72,73]. Briefly, subjects were sedated and paralysed by administration of 1:2:7 %v/v xylazine: Ketamine: deionised water, 100 μL, intraperitoneal (IP) injection, once. Mechanical shears were used to remove hair from a 2 cm² body surface area overlying the posterior hindlimb. Sterile 70% v/v Ethanol: deionised water swabs were used to remove disinfect the skin surface. Tumour cell suspensions containing 100,000 cells in 100 μL were loaded into syringes with permanent 28 G needles. Needles were inserted into the skin to the level of the dermis and the full volume was injected. Subjects were returned to the enclosure and monitored for complete recovery from anaesthesia. Humane endpoints for subject withdrawal were as follows: tumour diameter ≥15 mm, ulceration or body condition score <3 (ordinal scale).

## RMA-S MHC-I thermostability assay

H-2 stabilization experiments were performed adhering to standard protocol[74]. RMA-S cells were placed in culture at 25 °C 5% CO₂ for 18 h, then incubated peptides at the stated concentration for 30 min. at 30 °C 5% CO₂, then incubated for 3 h at 37 °C 5% CO₂. Cells were then washed twice with PBS, stained with fluorophore-conjugated monoclonal antibodies specific to H2-Kᵇ (Clone: AF6-885, Biolegend) or H-2Dᵇ (Clone: KH95, Biolegend) [0.5 μg/mL], 4 °C, 30 min, washed twice with PBS, fixed with PFA 1% w/v. Data were acquired on BD LSRFortessa.

## MHC tetramer staining

MHC tetramer staining was performed[75]. MHC tetramer reagents were non-covalently linked to streptavidin-PE (Invitrogen, cat#: S866) and/or streptavidin-APC (Invitrogen, cat#: S868) 1:1 mol:mol. Single-cell suspension of total PBMC/splenocytes or isolated CD8+ T cells were suspended in PBS (FCS 2% w/v, EDTA 2 mM) supplemented with dasatinib (SelleckChem, cat#: S1021) 50 nM, then incubated 30 min, 20 °C. MHC-tetramer (100 nM), and anti-mouse CD8a (2.5 × 10⁻⁴) g/L, Clone: CT-CD8a) was added then incubated, 60 min, 4 °C. The cells were then washed with PBS (FCS 2%, EDTA 2 mM) twice, then suspended in PBS (Paraformaldehyde, 1% w/v) and stored at 4 °C until use.

## Antigen-specific T Cell Clones and TCR Sequencing

CpG-C/ODN-2395 (₅T*C*G*T*C*G*T*T*T*T*C*G*G*C*G*C*G*C*G*C*C*G) was produced by IDT. Subject animals were treated with vaccines as described above. Six days post-immunization a single-cell suspension of MutuDC cells 1 × 10⁵ cells, 100 μL, (1 × 10⁹ cells/L) was added to round-bottom 96-well microtiter plate wells, then incubated for 37 °C, 24 h. IMDM FCS 8% v/v, supplemented with peptide 2 × 10⁻⁶ M, CpG-C/ODN-2395 and recombinant murine IFNγ (PeproTech, Cat#: 315-05) 1 × 10⁵ U/L were added, then incubated at 37 °C, 3 h. MutuDC cells were irradiated to a final dose of 50 Gy, then culture media was changed to RPMI FCS 10%v/v, 1 × 10⁻³ L, 1 × 10⁶ cells (1 × 10⁹ cells/L) supplemented with 2-mercaptoethanol (5 × 10⁻⁵ M), recombinant human IL-2 (Pepro-Tech, Cat#: 200-02) 2 × 10⁵ IU/L, recombinant murine IL-7 (PeproTech, Cat#: 217-17) 1 × 10⁻⁵ g/L and recombinant murine IL-15 1 × 10⁻⁵ g/L (PeproTech, Cat#: 210-015).

Seven days post-immunization CD8+ T cells were isolated from total splenocytes by bead-based affinity chromatography (Miltenyi, Cat#: 130-104-075). MHC tetramer staining was performed as described above. Single tetramer+ T cells were sorted onto peptide-pulsed irradiated MutuDC feeder-layers using a FACS Aria III (BD), then incubated at 37 °C, for 5–7d. Single-cell cultures were periodically visually assessed by bright-field microscopy for viability and cell expansion. Wells demonstrating secondary expansion were split, as necessary to maintain cell concentration (1 × 10⁶ cells/well). MHC tetramer staining was performed on expanded clonal T cell lines to confirm antigen-specificity.

Total RNA was isolated using RNeasy Micro Kit (Qiagen, cat#: 74034) according to manufacturer specifications. Paired 5′ RACE TCR sequencing libraries were generated using SMARTer Mouse TCR a/b Profiling Kit (Takara, cat#: 634403), according to manufacturer specifications. Library insert size was determined by Bioanalyzer DNA 1000 Kit (Agilent, cat#: 5067-1504). Library sequencing was performed using MiSeq Sequencer (Illumina, CA), 300 bp, paired-end, with targeted sequencing depth of 2 × 10⁷ reads/library. Demultiplexed FASTQ files were assembled into full-length TCR cDNA sequences using MiXCR[76].

## Flow cytometry staining

For experiments assessing IFNγ production by T cells in response to peptide stimulation, prior to flow cytometry, lymphocytes were cultured in RPMI (FCS 10%v/v) supplemented with anti-CD28 (Clone 37.51;

BioXCell, cat# #BE0015-1); peptide, 2.5 μM; GolgiPlug (BD, cat#: BDB555029) $1 \times 10^{-3}$g/L; GolgiStop, cat#: 554724) $1 \times 10^{-6}$M; for 6 h at 37 °C. Thereafter, and also for all independent flow cytometry experiments, flow cytometry staining for surface markers was performed by washing cells in phosphate-buffered saline (PBS), pH 7.4, and adding the following in PBS: fluorophore-conjugated antibodies anti-CD3 (17A2, BioLegend, cat# 100228, 1:200 dilution), anti-CD8 (Clone 53-6.7, BD Biosciences, cat# 564459, 1:500 dilution or Clone CT-CD8a, ThermoFisher/Invitrogen, cat # MA5-17597, 1:500 dilution), CD90.1 (Clone OX-7, Biolegend, cat# 202526, 1:500 dilution), CD44 (Clone IM7, Biolegend, cat# 103044, 1:500 dilution), and/or TCRβ (Clone H57-597, Biolegend, cat# 109222, 1:500 dilution) along with LIVE/DEAD fixable viability dye (ThermoFisher, cat# L23105, 1:1000 dilution) for 30 min at 20 °C. Then, intracellular cytokine staining was performed as in accordance with standard protocol, when applicable[77]. Cells were washed in PBS, and suspended in Fix/Perm solution (BD, Cat#: 554715) then incubated for, 30 min, 4 °C. The cells were washed twice in Perm/Wash solution (BD, Cat#: 554715), suspended in Perm/Wash solution containing anti-IFNγ (Clone: XMG1.2, Biolegend, cat# 505808, 1:800 dilution), then incubated, 30 min, 20 °C. Cells were then washed twice and suspended in PBS (PFA 1%w/v) and stored at 4 °C until use.

For experiments involving co-culture of B16F10 and T cells, B16F10 cell tumour cell suspensions consisting of $1 \times 10^5$ cells ($1 \times 10^9$ cells/L) $1 \times 10^{-5}$L DMEM FCS 10%v/v were added to 96-well flat bottom plates, then incubated 37 °C, 8 h. $1 \times 10^{-5}$L DMEM FCS 10%v/v supplemented with recombinant murine rmIFNγ $5 \times 10^4$ U/L was added, then incubated 37 °C, 12–16 h. Media was replaced with single-cell suspension of CD8+ T cells (1:1 ratio T cell:B16F10) in RPMI FCS 10%v/v, supplemented with anti-CD28 (Clone 37.51; BioXCell), then incubated 6 h at 37 °C. T cells were removed with gentle pipetting, then washed twice with PBS FCS 2%v/v, EDTA 2 mM before use in flow cytometry protocol above.

### Retrovirus plasmids for T cell transduction
pEF-ENTR A (696-6), pLenti X1 Zeo DEST (668-1), pLenti X1 Puro DEST (694-6), and pLenti X1 Zeo DEST (668-1) were gift from Eric Campeau & Paul Kaufman (Addgene# 17427, 17299 and 17297). pMSCV-IRES-GFP II (pMIG II) was a gift from Dario Vignali (Addgene# 52107). MSCV-IRES-Thy1.1 DEST was a gift from Anjana Rao (Addgene# 17442). pCL-Eco was a gift from Inder Verma (Addgene# 12371).

pMSCV(v5) γ-retrovirus transfer plasmids were constructed based on pMIG II, with the following modifications. pMIG II was linearised with EcoRI-HF (NEB, Cat#: R3101S), and AgeI-HF (NEB, Cat#: R3552S). The Woodchuck Post-transcriptional Regulatory Element (WPRE) from pLenti X1 Puro DEST (694-6), as well as the CD90.1/Thy1.1 from MSCV-IRES-Thy1.1 DEST were amplified by PCR. The complete cDNA for TCRζ, a furin (R/Arg-A/Ala-K/Lys-R/Arg) cleavage target and the Thosea asigna virus 2 A (T2A), followed by the complete cDNA for TCRβ, a furin cleavage target and porcine teschovirus-1 2A (P2A) were synthesised (Genscript, NJ)[78]. Point mutations in TRAC [p.T48C] and TRBC1/2 [p.S57C] were introduced to promote receptor pairing[79,80]. Segments were assembled in series into the linearised pMIG II backbone by flanking homology (NEBuilder HiFi DNA Assembly, NEB). Sanger sequencing (Genscript, NJ) was used to verify the correct sequence, order, and orientation of all constructed plasmids.

### Lentivirus plasmids for creation of tumour antigen-overexpressing tumour lines
Antigen-overexpressing tumour lines were engineered to normalise antigen levels; notably, Hsf2 is expressed at lower levels than non-mutated tumour antigens studied[81]. pENTR Gateway DONOR plasmids were constructed as follows. pEF-ENTR-A (696-6) was linearised with BamHI (NEB, Cat#: R3136S) and EcoRI-HF (NEB, Cat#: R3101S). The cDNA for mTagBFP2; G/Gly-S/Ser-G/Gly spacer; T2A; followed by the

cDNA corresponding to the 25 amino acid segments (Supplementary Table 3) surrounding indicated neoantigen peptides and a c-terminal flag (-DYKDDDDK) tag were synthesised (Genscript, NJ). Segments were assembled in series into the linearised pENTR-A backbone using the flanking homology method (NEBuilder HiFi DNA Assembly, NEB). pLenti transfer vectors were generated using LR Clonase II (Thermo-Fisher, Cat#: 11791020). Sanger sequencing (Genscript, NJ) was used to verify the correct sequence, order, and orientation of all constructed plasmids.

### Viral vector production
Ecotropic γ-retrovirus (γRV) particles were produced by transient co-transfection of HEK293T Platinum-Eco (Platinum-E) cells in accordance with standard protocol[82]. Platinum-E cells were seeded in 6-well microtiter plate in $1.5 \times 10^{-3}$L DMEM FCS 10%v/v, $1.2 \times 10^6$ cells ($1.27 \times 10^5$ cells/cm²) then incubated at 37 °C, 36 h. Culture media was replaced with DMEM FCS 10%v/v omitting Penicillin-Streptomycin then incubated at 37 °C, 60 min. Transfection particles were prepared by mixing MSCV-based γRV transfer vector (pMSCV), and packaging vector (pCL-Eco) 2:1 mol/mol ratio with FuGENE 6 lipid transfection reagent (Lonza, Cat#: E2691) according to manufacturer specifications. Transfection particles were added dropwise to Platinum-E cells, then incubated at 37 °C, 12 h. Culture media replaced with DMEM (FCS 10%v/v, HEPES 20 mM, GlutaMAX), then incubated at 37 °C, 36 h. Viral supernatant was collected at 48 h and 72 h post-transfection and centrifuged at $1000 \times g$, 5 min before use.

### Retroviral transduction
γRV transduction of CD8+ T cells was performed[83]. First, 250 μL PBS containing Retronectin/rFN-CH) (Takara, Cat#: T100B) 2e-2g/L, was added to non-treated 24-well tissue culture dish, then incubated at 4 °C, 12 h. Retronectin solution was removed, and 500 μL PBS containing BSA 2%w/v was added, then incubated 20 °C, 30 min. BSA solution was removed, and 1 mL viral supernatant was added. The plate was sealed, then centrifuged $2000 \times g$, 2 h. CD8+ T cells, activated for 24 h were centrifuged $500 \times g$, 5 m then suspended RPMI FCS 10%v/v, 1 mL, $1 \times 10^6$ cells ($1 \times 10^9$ cells/L) supplemented with 2-mercaptoethanol ($5 \times 10^{-5}$M), rhIL-2 $2 \times 10^5$ IU/L. Viral supernatant was removed and the cell suspension was added, centrifuged $2000 \times g$, 2 h, then incubated at 37 °C, 24 h. Transduction was performed twice, 24 h and 48 h post-activation.

### Lentivirus transduction
B16F10 cells were cultured in DMEM FCS 10%v/v to 75% confluence. TrypLE (ThermoFisher, cat#: 12605010) was added, then incubated at 37 °C, 10 min. Cells were centrifuged, $500 \times g$, 5 min then re-suspended in $1.5 \times 10^{-3}$ L DMEM FCS 10%v/v, $1 \times 10^9$ cells/L and transferred to a 6-well microtiter dish. Lentivirus particles were added to $1.5 \times 10^{-5}$ L DMEM FCS 10%v/v containing polybrene (EMDMillipore, cat#: TR-1003-G), $1 \times 10^{-1}$ g/L. Lentivirus suspension was added to B16F10 cells, then incubated at 37 °C, 24 h. Lentivirus supernatant was removed, and replaced with DMEM FCS 10%v/v, then incubated at 37 °C, 24 h. Lentivirus transduction was determined 48 h post-transduction by flow cytometry. Uniform populations of B16F10 mTagBFP2-minigene cells were isolated by fluorescence-activated cell sorting using a FACSAria III (BD Biosciences). Reference cell banks were generated on sort completion.

### CRISPR:spCas9 RNP electroporation
CRISPR: Cas9 Ribonucleoprotein transduction of naïve CD8+ T cells was performed in accordance with established protocol[84], here to knockdown TCRα and TCRβ. CD8+ T cells were isolated from preparations of total splenocytes by negative selection using magnetic isolation beads (Miltenyi, Cat#: 130-104-075), according to manufacturer specifications. Cells were suspended $1 \times 10^9$ cells/L in RPMI

('RPMI FCS 10%v/v', HEPES 20 mM, GlutaMAX, Pyruvate, Non-essential amino acids, Penicillin-Streptomycin), supplemented with 2-mercaptoethanol (Gibco, cat#: 31350-010) $5 \times 10^{-5}$ M, recombinant murine IL-7 ('rmIL7', PeproTech, Cat#: 217-17) $1 \times 10^{-5}$ g/L, then incubated at 37 °C, 12 h.

Antibody-coated plates were prepared as follows. $2.5 \times 10^{-4}$ L PBS solution containing monoclonal antibodies specific containing anti-CD3 (BioXCell Cat#: BE0001-1) $1 \times 10^{-4}$ g/L, and anti-CD28 (BioXCell Cat#: BE0001-1) $5 \times 10^{-5}$ g/L was added to each well of a 24-well tissue culture dish, then incubated at 4 °C for 12 h.

CRISPR:Cas9 Ribonucleoproteins (RNPs) were produced as follows. Synthetic CRISPR RNA (crRNA) and transactivating RNA (tracrRNA, cat#: 1072532) were purchased from IDT. Customised crRNA sequences are as follows: TRAC, 5'-TCTGGGTTCTGGA TGTCTGT PAM: GGG, and TRBC1/2, GTCACATTTCTCAGATCCTC PAM: TGG. Duplex crRNA:tracrRNA was produced according to the manufacturer's specification, aliquoted, and stored at −80 °C. RNP were produced by combining duplex RNA and TrueCut Cas9 Protein v2 (ThermoFisher, Cat#: A36498), $1.5 \times 10^{-12}$ mol:$5 \times 10^{-12}$ mol, then incubating at 20 °C, 10 min.

CD8$^+$ T cells were washed twice with PBS then suspended in P4 Nucleofector solution (Lonza, Cat#: V4XP-4032) $2 \times 10^{-5}$ L, $1 \times 10^{-5}$ cells ($5 \times 10^{11}$ cells/L). Alt-R Cas9 Electroporation Enhancer $1 \times 10^{-6}$ L, $1 \times 10^{-4}$ M ($4 \times 10^{-6}$ M) was added, followed by CRISPR:Cas9 RNPs. The cells were then transferred to 4D-Nucleofector X Unit (Lonza, Cat#: V4XP-4032). RNP were delivered by electroporation using 4D-Nucleofector (Lonza, cat#: AAF-1002X), pulse code: DS-137. The cells were then carefully distributed into a 96-well round-bottom tissue culture dish, containing RPMI FCS 10%v/v $2 \times 10^{-4}$ L, $2 \times 10^6$ cells/well ($1 \times 10^{10}$ cells/L), then incubated 37 °C, 2 h.

CD8$^+$ T cells were transferred to RPMI FCS 10%v/v, $5 \times 10^{-4}$ L, $1 \times 10^6$ cells ($2 \times 10^9$ cells/L) supplemented with 2-mercaptoethanol ($5 \times 10^{-5}$ M), recombinant human IL-2 ('rhIL-2', PeproTech, cat#: 200-02) $2 \times 10^5$ IU/L and recombinant murine IL-12p70 (PeptoTech, cat#: 210-12) $1 \times 10^{-5}$ g/L, then plated in 24-well plates coated with CD3/CD28 and incubated at 37 °C, 24 h.

crRNA targeting TRAC and TRBC1/2 were designed with the IDT CRISPR-Cas9 guide RNA server (IDT), using reference genomic sequence of the TRBC1 (GRCm38.p6 C57BL/6 J, ch6:41537984-41538423), TRBC2 (GRCm38.p6 C57BL/6 J, ch1441546489-41547115), and TRAC loci (GRCm38.p6 C57BL/6 J, ch14: 54219921-54224806) were used as target sequence references. crRNA was selected if PAM and/or crRNA nt p1-5 crossed an exon boundary. crRNA validation was performed by flow cytometry 72 h post electroporation. CD8$^+$ T cells exposed to RNP targeting TRAC, TRBC1/2 or were stained with CD3e-BV421 (17A2, Biolegend, cat#: 100228) CD8-FITC (53-6.7, Biolegend, Cat#: 100706), and TCRβ-PE (H57-597, Biolegend, Cat#: 109222), and isotype controls. crRNA achieving >90% reduction in surface CD3e/ TCRb were retained. RNP-treated cells were assessed following transduction with γRV TCR-47BE7 and stained with H2-D$^b$/Hsf2(47) pMHC tetramer. crRNA achieving >90% transduction, with surface expression of tgTCR determined by pMHC tetramer staining were retained.

## CRISPR/Cas9 RNP lipofection and knockdown cell isolation
B16F10 cells were cultured in DMEM FCS 10%v/v to 75% confluence. TyrpLE (ThermoFisher, cat#: 12605010) was added, then incubated at 37 °C, 10 min. Cells were centrifuged, $500 \times g$, 5 min then re-suspended in $1.5 \times 10^{-3}$ L DMEM FCS 10%v/v, $6.67 \times 10^7$ cells/L and transferred to a 6-well microtiter dish. CRISPR:Cas9 RNP were prepared as described above, with the following modifications. RNP transfection particles were prepared using Lipofectamine RNAiMax lipid transfection reagent (ThermoFisher, cat#: 13778100), according to the manufacturer's specifications. Transfection particles were added to $1.5 \times 10^{-3}$ L DMEM FCS 10%v/v, supplemented with polybrene

(EMDMillipore, cat#: TR-1003-G), then added to 6-well microtiter dish wells containing B16F10 single-cell suspensions. The combined mixture was incubated on an orbital shaker, 50 rpm, 5 min, 20 °C; then moved to incubate at 37 °C, 5%CO$_2$, 48 h. Treated B16F10 cells were then split into two wells, containing 1.5 mL DMEM FCS 10%v/v and maintained in culture to 75% confluence. Knockout of crRNA target gene products was measured by flow cytometry. Briefly, $1 \times 10^{-5}$ L DMEM FCS 10%v/v supplemented with recombinant murine IFNγ ('rmIFNγ', PeproTech, Cat#: 315-05) $5 \times 10^4$ U/L was added to cultures, then incubated 37 °C, 12–16 h. B16F10 cells were disassociated from the culture dish by adding cold PBS containing EDTA 5 mM, incubation at 20 °C 5 min, then gentle pipetting. Single-cell B16F10 suspensions were washed twice with PBS BSA 2%v/v, then PBS containing anti-H2-K$^b$ (Clone: AF6-88.5, Biolegend) and anti-H2-D$^b$ (Clone: KH95, Biolegend) or appropriate isotype control antibodies were added then incubated 4 °C, 30 min. Data acquisition was performed on LSR Fortessa (BD Biosciences). Target gene product surface protein expression level was defined using mock RNP-treated (positive control), and isotype control antibody (negative control) treated samples. To generate B16F10 H2-K$^b$ $^{-/-}$ and H2-D$^{b-/-}$ cell lines RNP-treated B16F10 cells were sorted with FACSAria III (BDBiosciences), re-verified by flow cytometry before being cryopreserved.

## Expression, folding and purification of H2-D$^b$/YGFRNVVHI
Soluble peptide-MHC monomers were synthesised in accordance with published methods[85]. Accordingly, the extracellular domain of H2-K$^b$, H2-D$^b$, as well as full-length human β2M (hβ2M) were cloned into pET3A plasmids (provided by Ton Schumacher, Netherlands Cancer Institute-Antoni van Leeuwenhoek (NKI-AVL)). For MHC tetramer production H2-K$^b$ and H2-D$^b$ expression constructs included a c-terminal biotin acceptor peptide (BirA-tag). Proteins were expressed in E. coli BL21(DE3) PLysS as inclusions bodies (IBs). IBs were solubilised in 100 mM Tris-HCl, pH 8.0, supplemented with urea 8 M, DTT 100 μM, and EDTA 10 mM. Insoluble material was cleared by centrifugation $20,000 \times g$, 20 min. Denatured proteins were added in a H2-D$^b$:hβ2M 1:2 mol:mol ratio to solution of Tris-HCl 100 mM pH 8.0, supplemented with L-arginine 400 mM, L-glutathione 500 μM, oxidised L-glutathione 50 μM, and EDTA 2 mM, PMSF 10 mM, protease inhibitor cocktail (Roche, Cat#:11873580001) and peptide 1.2 mM. The mixture was stirred at 4 °C for 72 h, and precipitate cleared by centrifugation at $20,000 \times g$, x 10 min, concentrated by centrifugal ultrafiltration using a 15 kDa membrane (Millipore, cat#: UFC903024) and sequentially purified by size exclusion chromatography (Superdex 75, GE Healthcare), followed by anion exchange (HiTrap Q HP 5, GE Healthcare). MHC monomers were concentrated by centrifugal ultrafiltration using a 4 kDa membrane (Millipore, cat#: UFC8030) and exchanged into HBS (HEPES 10 mM, NaCl 150 mM). Peptide-MHC protein complex identity was verified by polyacrylamide gel electrophoresis, then split into single-use aliquots before snap-freezing in liquid nitrogen, then stored at −80 °C until use. For MHC tetramer production site-specific enzymatic biotinylation by BirA biotin ligase was performed before the anion-exchange chromatography previously described.

## Expression, folding and purification of TCR 47BE7
Soluble TCR was synthesised following standard protocol[56]. Briefly, αβTCR extracellular domains were ordered as codon-optimised synthetic DNA fragments. A stabilizing disulfide bond was engineered by introducing point mutations into the TCR constant domains [TRAC p.T48C, TRBC1 p.S57C] and a C-terminal Gly-Ser 6x histidine tag was appended to TRBC1. DNA fragments were cloned into linearised pET28a plasmids by Gibson assembly, then transformed into E. coli BL21(DE3). Transformed cells were grown in Luria Broth (LB) for 4–6 h at 37 °C shaking at 200 rpm, induced with IPTG 100 mM, then

incubated for 4 h at 37 °C shaking at 200 rpm. Cells were pelleted by centrifugation at $4000 \times g$, resuspended in lysis buffer (Tris-HCl 50 mM, pH 8.0, sucrose 25%w/v, Triton-X100 1%v/v, EDTA 1 mM, lysozyme (Sigma-Aldrich, Cat#: L6876) and DNAse (Roche, Cat#: 10104159001). Protein IBs were sequentially washed (Tris-HCl 20 μM, pH 8.0, NaCl 150 mM, Triton-X100 0.5% v/v, EDTA 1 mM, DTT 1 mM) then (Tris-HCl 20 μM, pH 8.0, NaCl 150 mM, EDTA 2 mM, DTT 1 mM), then solubilised (Tris-HCl 100 μM, pH 8.0, Gdm-HCl 6 M, DTT 10 mM, EDTA 10 mM). Denatured protein was added dropwise to a final 1:1 molar ratio (TCRα:TCRβ) to a folding solution (Tris-HCl 100 mM, pH 8.0, urea 5 M, L-Arg 400 mM, L-GSH 5 mM, L-GSSG 500 μM, EDTA 2 mM, PMSF 10 mM). The mixture was stirred at 4 °C for 72 h, then cleared of precipitate by centrifugation at $20,000 \times g$, x10 min. Then dialyzed (Tris-HCl 10 mM, pH 8.0), using a 10 kDa MWCO membrane (Millipore, #Cat UFC8010) for 48 h, changing dialysate every 12 h. Proteins were concentrated by ultrafiltration using a 10 kDa MWCO membrane (Millipore, cat#: UFC903024) and purified by IMAC (HisTrap HP, GE Healthcare), then SEC (Superdex 75, GE Healthcare). Monodisperse fractions of appropriate molecular weight were concentrated and exchanged by ultrafiltration using a 4 kDa membrane (Millipore, Cat#: UFC8030) into HBS (HEPES 10 mM, NaCl 150 mM), then snap-frozen in liquid nitrogen and stored at −80 °C until use.

### Biolayer interferometry
The AVI-tagged version[86] of recombinant H2-D$^b$ was used to refold the pMHC in complex with YGFRNVVHI peptide; this was biotinylated using a BirA biotinylation kit (Avidity) and purified by gel filtration. The gel-shift assay using streptavidin (Fisher Scientific) verified protein biotinylation both after biotinylation and immediately before starting the BLI experiment.

Binding between pMHC and TCR was measured using a BLI Octet RED96 instrument and streptavidin-conjugated SA chips in accordance with the manufacturer's (Pall ForteBio, Menlo Park, CA) protocol. For all analyses, HST buffer (10 mM HEPES, pH 7.5, 100 mM NaCl, 0.01% Tween-20, 1% BSA) was used. Biotinylated pMHC was diluted to 2–10 μg/mL, immobilised on a sensor chip, and equilibrated in HST buffer for 1 min. Following this step was TCR association (60 s) in 200 μL of TCR solution (concentrations ranging between 0 μM and 20 μM), and then dissociation in 200 μL of HST buffer for 3–20 min. Assays were conducted at 25 °C. The Octet® 9.1 System Data Analysis software was employed to determine dissociation constant values from these data.

### Protein crystallization and data collection
TCR 47BE7 and H2-D$^b$/$_{68}$YGFRNVVHI were refolded and purified separately as described above. Protein crystallization was performed by sitting drop vapour diffusion. 96-well INTELLI-plates (Art Robbins Instruments, Cat# 102-0001-20) were seeded with a Mosquito crystallization Robot (SPT Labtech) utilizing a 1:1 v/v protein to precipitant ratio then incubated at 18 °C until crystal formation. H2-D$^b$/YGFRNVVHI formed prism-shaped crystals (Tris-HCl 0.1 M, pH 8.5, sodium acetate 0.2 M, PEG3350 20–25%). Crystals were cryoprotected with the same mother liquor, supplemented with ethylene glycol 25% v/v then flash-frozen in liquid nitrogen and stored until data collection.

The ternary TCR 47BE7/H2-D$^b$/$_{68}$YGFRNVVHI complex was formed by mixing both proteins in a 1:1 molar ratio, concentrating the mixture to 8 mg/mL by ultrafiltration using a 30 kDa MWCO membrane (Millipore, Cat#: Z717185). The ternary complex formed rod-shaped crystals (Bis-Tris 0.1 M, pH 5.5, PEG3350 20–30%, lithium sulfate 0.2 M, glycerol 15% v/v). Crystals were directly flash-frozen in liquid nitrogen and stored until use.

### X-ray diffraction data collection, structure solution and refinement
X-ray diffraction data were collected at Argonne APS beamline 19BM using SBCCollect. Data were indexed, integrated, and scaled using HKL3000 and the AIMLESS/CCP4 programme suite[87] The crystal structure of H2-D$^b$/YGFRNVVHI was solved using PHASER with the reference search model (PDB: 5OPI) and refined using REFMAC and COOT 0.9.7[88]. The H2-D$^b$/YGFRNVVHI structure coordinates and structure factors are accessible via Protein Data Bank accession code (PDB: 7N9J). The structure of TCR 47BE7/H2-D$^b$/$_{68}$YGFRNVVHI ternary complex was solved using the binary complex as a search model, the 2FoFc map of which was used to sequentially build and refine the TCRα and TCRβ chains for TCR_47BE7 heterodimer by alternating refinement in REFMAC with model building and refinement in COOT. The 47BE7/H-2-Db/$_{68}$YGFRNVVHI ternary complex coordinates and structure factors are accessible via Protein Data Bank accession code (PDB: 7NA5). The X-ray diffraction data collection, processing and refinement statistics are presented in Supplementary Table 2.

Solvent exposed surface area (SASA, $A^2$) for individual amino acid residues was calculated using NACCESS software[89], and was expressed as a percentage of all residue surface area (ASA). The SigmaA-weighted 2Fo-Fc or Fo-Fc electron density maps in CCP4 format for drawing were generated in COOT 0.9.7 using amplitude (FWT) and phase (PHWT) data from the REFMAC mtz output structure factor file.

### Structural modelling
Structural modelling of wild-type H2-D$^b$/Hsf2$_{68-76}$ was performed using PepFlexDock in Rosetta. The high-resolution crystal structure of H2-D$^b$/Hsf2 p.K72N$_{68-76}$ served as the template, into which the position 5 asparagine to lysine mutation was introduced using COOT or the PyMol mutagenesis function. Energy minimization was then performed using FlexPepDock online server[90]. Each model was truncated to the H2-D$^b$ domain 1 (AA 1–180) and peptide. The top scoring models were compared in COOT, PyMol. Images were generated using PyMol and Microsoft Excel 2019.

## Data availability
Whole exome sequencing data and RNA sequencing data have been deposited on the Sequence Read Archive (SRA) available on the National Institutes for Health (NIH) National Centre for Biotechnology Information (NCBI) website, with the accession number PRJNA1049400. The H-2-D$^b$/hβ2M/$_{68}$YGFRNVVHI binary structure coordinates and structure factors are accessible via Protein Data Bank accession code 7N9J. The H-2-Db/hβ2M/$_{68}$YGFRNVVHI:47BE7 ternary complex coordinates and structure factors are accessible via Protein Data Bank accession code 7NA5. Source data are provided with this paper.

## Code availability
Peptide-MHC-I binding prediction tool NetMHCpan (v.4.1) software used in this study for neoantigen prediction is available online at https://github.com/openvax/vaxrank.

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

## Acknowledgements

Crystallographic data shown in this report are derived from work performed at Argonne National Laboratory, Structural Biology Centre at the Advanced Photon Source. SBC is operated by UChicago Argonne, LLC, for the U.S. Department of Energy, Office of Biological and Environmental Research under contract DE-AC02-06CH11357. This study was supported by NIGMS R01 GM124489, NCI R01 CA243486, NCI P50 CA225450 New York University Melanoma SPORE grant, NCI U01 CA214354, Mark Foundation for Cancer Research, Merck Oncology Translational Programme Concept 59776, and P30 CA016087 Cancer Centre Support Grant all awarded to M.K., and the Melanoma Research Alliance, the Kimberly and Eric Waldman fund (Mount Sinai Department of Dermatology), and NCI R01 CA201189, all awarded to N.B.; J.N. was supported by Cancer Research Institute Lloyd J. Old Memorial Post-doctoral Fellowship in Tumour Immunology, T32AI078892-12 and T32CA078207-21. The authors would like to thank 2021 Bhardwaj Laboratory summer undergraduate student Casey Lutnick for helping with reconstituting peptides and the members of the Bhardwaj and Krogsgaard laboratories for many discussions.

## Author contributions

Conceptualization: N.B. and J.P.F.; Methodology: J.P.F., G.M.L., A.S.I., J.H.N., Y.P., L.P. and R.A.S.; Investigation: J.P.F., J.H.N., Y.P. and L.P.; Visualization: J.P.F., J.H.N. and Y.P.; Funding acquisition: N.B. and M.K.; Project administration: N.B. and M.K.; Supervision: N.B. and M.K.; Writing

– original draft: J.P.F., J.H.N. and Y.P.; Writing – review & editing: N.B., M.K., J.P.F., J.H.N. and Y.P.

## Competing interests

J.P.F., J.N., Y.P., L.P. and R.A.S. declare no competing interests. A.S.I. and G.M.L. are employees and shareholders of Barinthus Therapeutics, which is commercializing cancer vaccines, including SNAPvaxTM. MK serves on the scientific advisory board of NexImmune, Genentech and Merck and Co. and received research support from Merck Sharp & Dohme Corp., a subsidiary of Merck and Co., Inc., Genentech, Biogen, Novartis and the Mark Foundation. N.B. serves as an advisor/board member for Apricity, BreakBio, Carisma Therapeutics, CureVac, Genotwin, Novartis, Primevax, Rome Therapeutics, Tempest Therapeutics, and as a consultant for Genentech and ATP. N.B. receives research support for Dragonfly Therapeutics, Inc., Harbour Biomed Sciences, and Regeneron Pharmaceuticals, Inc. N.B. serves on the Scientific Advisory Council/Board of the Cancer Research Institute, Duke University CHAVD, MD Anderson Cancer Centre, Parker Institute for Cancer Immunotherapy, and the American Association for Cancer Research. N.B. serves on the Grants/Research support council for the Cancer Research Institute, Ludwig Institute for Cancer Research, Melanoma Research Alliance, Leukaemia and Lymphoma Society, Pershing Square Sohn Cancer Research Alliance, and Stand Up to Cancer.
