## [Peer Review File · Nature Communications]

Reviewers' Comments:

Reviewer #1:

Remarks to the Author:

This manuscript discusses a well conducted experiment that has paid attention to detail. The quality of the results are excellent. Some computer aided modelling was used to extend the results and make predictions or extrapolations that are well informed, but the conclusions drawn are not safe enough, in my opinion. These will need some attention, and perhaps modification in the light of my comments below. There are also a few editorial corrections required. My recommendation is that the manuscript should be accepted for publication, if the editor is satisfied that my suggestions for changes are met.

Notes for the authors - Please note that in the pdf document I got, the main text starts at page 3.
=====

- Abstract: shared viral immunogenicity is topical, but might be a distraction. The main text only skims this like an afterthought, and does not have a serious discussion around this point.

- Introduction, p 4: Would the authors elaborate on the reason for restricting the investigation to 7 neoAg?

- p 5, para 2, 'used to organ neoAg into two principal classes': Presumably 'organ' should be 'organise'. I prefer the use of 'groups' instead of 'classes' to avoid confusion with Class I/II MHC. This is very relevant at the bottom of p5: 'prototype murine class II neoAg' and further, particularly in the discussion section towards the end of the text.

- p 5, last para: The discussion comparing pK5 and pN5 is shielding the reader from an important point, relying on the credibility of in-silico modelling. I have no problem with that at all. In this case though, I have serious doubt that the conformation of the peptide would be the same when residue 5 is the wt type K instead of N as in this study. I assume that the in-silico modelling simply replaced the side chain of Asn as determined from the experiment, by coordinates for a Lys, before running a molecular dynamics minimisation. This indeed would show Asn to be much more favourable, as its side chain perfectly complements the side chain of Gln97, both in size and charge complementarity, a perfect fit. But altering the side chain to be a Lys immediately creates a clash with Gln97, because its side chain is too long. It has to bend away from this side chain. Being a charged residue, it can only move towards Glu9, a little too far, or Gln70, still not a perfect fit. In my opinion, this side chain would instead point outwards from the groove, like pR4. Many other peptides are presented thus, with all central residues pointing outwards, what I describe as the OMEGA peptide conformation, as opposed to the M conformation as in this case. This would make perfect sense, and would also explain the failure of the wt peptide to activate the identified TCR. If anything, this could be a perfect demonstration of the basis for the thymus to eliminate any TCR that would recognise the wt peptide (self) while permitting TCRs that recognise the mutant. This conjecture can be settled by determining the structure of the MHC presenting the wt peptide. If that structure had been determined, it is not being discussed in this manuscript. As such, the experiment is incomplete, making the conclusions a little unsafe. It may be an oversight, or the circumstances may have militated against determining that extra structure, and I would not insist on having it completed for the sake of the paper, but the authors may wish to re-examine their comments pursuing my notes.

- p 6: This page starts with a 12 line description of how peptides lie in the MHC groove. All very interesting, but presented as a hypothesis, later validated by structure determination. Being an experimentalist, I would normally determine the structure first and work out all this description later, with no need to hypothesise. However, I would accept that this was the genuine sequence of events, but I am just checking. If the structure determination did indeed precede the description, this para needs to be reworded please.

- p 6, 8 lines from the bottom: It is not at all surprising that the binding in pocket B is low affinity. pG2 does not have a side chain, therefore there is limited chance of interaction with the MHC. This pocket can take any side chain, and the general design of the MHC makes it imperative that the

small side chains would leave cavities. But these cavities are solvent accessible, at least before folding. So solvent molecules may get trapped in there, but that is incidental. It also allows the MHC to be a general presenter of antigens.

- p 6, last line: Residue 5 in the peptide is not normally an anchor residue, as I argue above. I leave it to the authors to consider rewording this statement. Indeed, on page 7, this residue is described a 'secondary anchor'!

- p 7, l 3: 'W73-Y147-W147' should be 'W73-W147-Y156'

- p 7, l 7: 'DFW' should be 'DWF', as defined 2 lines before. The argument that follows seems to emphasise that the stability of the peptide binding into the groove is a good thing. That notion leaves me cold. The TCR response is no respecter of peptide-MHC stability, otherwise we would not observe allergies triggered by self-recognition after incidental cross-reactivity (diabetes). Also, in one case, a 'stabilised' peptide by changing pA2 to pL2, activated a different TCR which did not recognise the wt peptide as well (melanoma); the correct activation was dependent on extracting the N-terminus of the peptide out of the MHC pocket upon engagement by the cognate TCR. The authors may want to change the tone of this para to shift away from an argument on the peptide stabilisation. A good response to the mutant neoAg requires a DIFFERENT TCR, which must not engage the wt peptide. The authors have identified one, and I congratulate them.

- p 7, last para: The 'p6-p8 ridge' is not unique. It occurs in the human NYESO-1 peptide recognised by 1G4, again propped up by a Trp side chain in the MHC. In that case, the one I am familiar with, residues 4 and 5 (MW) are pointing outwards.

- p 8, para 1, last line: 'confirmation' ==> 'conformation'

- p 8, para 2, l 2: 'G18-E19' should be 'E18-E19', as residue 18 is described as E 2 lines later. This is also its assignment in the deposited model.

- p 8, para 2, l 4: 'vis-a-vis the backbone C-alpha towards while' does not make sense. This sentence needs to be recast.

- p 10, para 2, last sentence: I disagree with this statement. Any side chain other than Asn at position 5 would not have the same conformation, because Asn makes specific contacts with the side chain of MHC residue Q97. The list of mutations tested cannot make that specific interaction. BLI may have detected recognition by TCR, but it cannot inform on peptide conformation. Size, shape and charge pattern are what matter.

- p 10, last para: Again, hypothesis does not necessarily match the facts. pK5 being an outlier is probably a case in point, should that structure be determined.

- p 11, para 1: At last, the authors recognise that pK5 does not fit in the pocket. The response, though, is surreal. They propose that the binding pocket needs to reorganise itself! They propose rotating the side chain of W73 by 180 deg, and cause wholesale rearrangement in the pocket, just to keep the K5 side chain tucked in. No, it's the peptide that would adopt a new conformation. The MHC fold is very stable, while the peptide is malleable. The chase after a conserved main chain for the peptide is like barking up the wrong tree. This conclusion is definitely wrong. It is not accidental that the pK5 variant is the outlier in the energy interaction calculations and measurements.

- Fig 1 legend: What is BA-Rank?

- Fig 2, legend: Again, ACT not defined

- Fig 3, panel c: pN5 is shown contacting residue 'N97' which is actually 'Q97'

- Fig 3, panel f and its legend: The thermal parameters are referred to with the commonly used B-factor. The discussion in the main text on page 7 calls them DWF. The authors should use one style

of reference to this quantity.

Reviewer #2:

Remarks to the Author:

Finnigan et al. focused on the specificity and cross-reactivity of T cells towards neoantigens (neoAgs) and performed a wide range of experimental methods including RNA-seq, x-ray crystallography, in silico study, peptide library, and in vivo evaluations. The authors successfully identified neoAgs of B16F10, a common murine implantable tumor model of melanoma, using combined techniques. Furthermore, a high-affinity TCR targeting H2-Db-restricted neoAg, Hsf2K72N, was identified. This TCR, TCR 47BE7, showed specific recognition of neoAg, Hsf2K72N of B16F10 in vitro and in vivo. Crystal structure of the peptide-MHC (pMHC, neoAg, Hsf2K72N/H2-Db) revealed that the K72N mutation makes new interaction with Asn at the bottom of the MHC groove (but not to TCR directly, categorized as class II), inducing C-terminal site of the peptide (pV6-pH8) rigid to form a solvent-exposed hydrophobic arch in H2-Db. Moreover, the crystal structure of the peptide-MHC-TCR complex (neoAg, Hsf2K72N/H2-Db, TCR 47BE7) complex showed that TCR 47BE7 mainly recognized this C-terminal site. In addition, this recognition is similar to that of an influenza peptide-H2-Db, suggesting common structural features between neoantigens and viral peptides. The study is extensive and interesting. I have some comments as follows.

- 1) The SPR analysis for TCR 47BE7 binding to Hsf2WT/H2-Db is helpful for understanding the functional difference between wild-type and K72N mutant quantitatively, even though Fig. 4g showed that WT peptide did not exhibit IFN γ production.
- 2) Fig. 2b-d; only the pM order of the peptides can induce IFN γ production of neoantigen-reactive T cells, even though the KD of TCR-pMHC binding is μ M level. Did transgenic T cells significantly express TCRs?
- 3) Please add a figure showing a structural comparison of the C-terminal peptide region between this neoAg and an influenza peptide-H2-Db for readers to easily understand.
- 4) Fig. 3c; The side chain of pN5 is red-based colored. However, oxygens are also colored red and thus a bit difficult to discriminate.
- 5) Fig. 4c; Two CDR1 β s exist...

Reviewer #3:

Remarks to the Author:

The manuscript by Finnigan and colleagues examines the B16F10 murine tumor and isolates several candidate mutated peptides, also called neoantigens, that are expressed by the tumor. In particular, they focus on TCR 47BE7 which targets H2-Db-restricted neoAg Heat Shock Protein 2 (Hsf2 p.K72N68-76). With a series of biophysical measurements, the manuscript attempts to define the rules governing neoantigen immunogenicity with the anticipation of these rules being clinically relevant. Overall, the paper is well written, but suffers in novelty and (large portions) being descriptive. Major concerns are as follows;

- 1) The predominant T cell (TCR) clone chosen for modeling, 47BE7, does not recognize the native tumor, unless the antigen is over-expressed. This is not addressed and it severely impacts the rationale, especially since the peptide vaccination by itself is able to elicit some tumor control.
- 2) Figure 3 and 4 are largely descriptive and makes some predictions and correlations. However, none of these are truly tested and so the rules defining neoantigens, which the manuscript sets out to define, are left hanging in the balance.
- 3) Figure 2h,i needs ACT of tumor-nonspecific T cells as a control not vehicle. This experiments should have been repeated with all the candidate peptides/TCR identified in Fig2a. Tumor rejection is a far more important parameter clinically than simply measuring IFN γ
- 4) It is difficult to understand the clinical relevance of this study given that the TCR that is the primary focus of the study does not reject the parental tumor.
- 5) How universal the putative neoantigen-defining rules are to the next TCR and the next tumor neoantigen/MHC is unclear, or will these putative rules be specific to this one p/MHC/TCR?

Reviewer #4:

Remarks to the Author:

The authors provide a comprehensive analysis of neoantigen recognition in the murine B16F10 melanoma model, starting with exome sequencing to define epitopes and subsequent T cell cloning and TCR characterization. This is an impressive compilation of functional and molecular immunology that provides insight into tumor neoantigen recognition. The biophysics and structural biology is thorough and of high quality.

There is a significant amount of data generated and presented in this analysis, so understandably there are challenges in data presentation. However, the figures are currently very difficult to read and will be even more difficult to read when formatted for a publication. Perhaps moving some of the data to the supplement might help. The font size should also be increased so that the reader does not have to zoom in on a pdf, as this will make reading on a printout almost impossible. Other than this, the manuscript provides an abundance of interesting molecular and biophysical data that will add to our understanding of neoantigen recognition in the anti-tumor T cell mediated response.

REVIEWER COMMENTS & RESPONSES

We thank the reviewers for their comments and address these below in response to each reviewer's critiques.

We have made extensive changes throughout the manuscript to improve it in response to reviewers' comments, including making figures larger and more legible. We completely reworked the structural section of the manuscript, adding extensive modeling analyses based on published datasets and moving this data to the supplemental section, as suggested. We also edited our structural figures in the main text for better clarity. Our discussion section of the manuscript has been extensively modified and extended to fully capture the novelty of the study, data interpretation in the broader context of other published studies, and future directions. We believe these changes have vastly improved the quality of our manuscript and thank the reviewers for their thoughtful and helpful comments.

Reviewer #1 (Remarks to the Author):

This manuscript discusses a well conducted experiment that has paid attention to detail. The quality of the results are excellent. Some computer aided modelling was used to extend the results and make predictions or extrapolations that are well informed, but the conclusions drawn are not safe enough, in my opinion. These will need some attention, and perhaps modification in the light of my comments below. There are also a few editorial corrections required. My recommendation is that the manuscript should be accepted for publication, if the editor is satisfied that my suggestions for changes are met.

Notes for the authors - Please note that in the pdf document I got, the main text starts at page 3.

=====

We thank reviewer #1 for their thoughtful comments and recognition of the quality of results. We respond below to each critique and believe we have addressed each issue.

- Abstract: shared viral immunogenicity is topical, but might be a distraction. The main text only skims this like an afterthought, and does not have a serious discussion around this point.

We agree and have removed this concept from the abstract and significant conjectures from the discussion as well. We recognize that the shared motif is a feature that has been observed for H2-D^b-peptide complexes and that this does not necessarily signify any likeness between influenza and melanoma antigens. Separate from this however is the notion that Hsf2 p.K72N behaves as a "non-self" peptide because it is selectively recognized, with little cross-reactivity to the wild type peptide, by its cognate TCR, 47BE7.

- Introduction, p 4: Would the authors elaborate on the reason for restricting the investigation to 7 neoAg?

We elaborated upon the selection of neoAg in the introduction. After computational prediction of neoantigen pMHC complexes, we vaccinated with 12 predicted long peptide neoAgs. H2-K^b or H2-D^b restriction and specific minimal peptide epitope determination was resolved using a panel of 50+ MHC tetramers; from this, we observed only seven MHC-I restricted pMHCs elicited T cell responses, hence the decision to restrict the following studies to 7 neoantigens (Fig. 1c,d and Extended Data Fig. 1).

- p 5, para 2, 'used to organ neoAg into two principal classes': Presumably 'organ' should be 'organise'. I prefer the use of 'groups' instead of 'classes' to avoid confusion with Class I/II MHC. This is very relevant at the bottom of p5: 'prototype murine class II neoAg' and further, particularly in the discussion section towards the end of the text.

All requested changes have been addressed in the revised text; "class" has now been replaced by "group" where relevant.

5. - p 5, last para: The discussion comparing pK5 and pN5 is shielding the reader from an important point, relying on the credibility of in-silico modelling. I have no problem with that at all. In this case though, I have serious doubt that the conformation of the peptide would be the same when residue 5 is the wt type K instead of N as in this study. I assume that the in-silico modelling simply replaced the side chain of Asn as determined from the experiment, by coordinates for a Lys, before running a molecular dynamics minimisation. This indeed would show Asn to be much more favourable, as its side chain perfectly complements the side chain of Gln97, both in size and charge complementarity, a perfect fit. But altering the side chain to be a Lys immediately creates a clash with Gln97, because its side chain is too long. It has to bend away from this side chain. Being a charged residue, it can only move towards Glu9, a little too far, or Gln70, still not a perfect fit. In my opinion, this side chain would instead point outwards from the groove, like pR4. Many other peptides are presented thus, with all central residues pointing outwards, what I describe as the OMEGA peptide conformation, as opposed to the M conformation as in this case. This would make perfect sense, and would also explain the failure of the wt peptide to activate the identified TCR. If anything, this could be a perfect demonstration of the basis for the thymus to eliminate any TCR that would recognise the wt peptide (self) while permitting TCRs that recognise the mutant. This conjecture can be settled by determining the structure of the MHC presenting the wt peptide. If that structure had been determined, it is not being discussed in this manuscript. As such, the experiment is incomplete, making the conclusions a little unsafe. It may be an oversight, or the circumstances may have militated against determining that extra structure, and I would not insist on having it completed for the sake of the paper, but the authors may wish to re-examine their comments pursuing my notes.

We thank the reviewer for these thoughtful comments and agree that while molecular modeling predicts possible conformations, it cannot be used to make conclusions. The modeling data have to be corroborated with the existing experimental data offering at least indirect support of the models. First, we performed a more detailed analysis of modeling data produced by the PepFlexDock server. Using the coordinates of the solved crystal structure between H2-D^b and Hsf2 p.K72N, we produced a series of models in which pN₅ of the epitope is replaced with other amino acid residues, including lysine. For each model two structural peptide variants were designed, in which the residue at P5 adopted either anchor (1) or non-anchor (2) orientations, respectively. The vast majority of the substitutions resulted in the amino acid residue at P5 adopting anchor conformations in the highest scoring solutions. Such modeling predictions were in a complete agreement with the available crystallographic data (see an example presented in Extended Data Fig. 6c-e). By contrast, there were two types of the top scored conformations generated between Hsf2-WT peptide (pK₅) and H2-D^b – both with similar docking scores. In model #1 the pK₅ side chain remained in the anchor position, but the H2-D^b W73 was displaced from the pocket E to accommodate the lysine side chain. This conformation was shown in the previous version of the manuscript. However, the alternate Hsf2_WT conformation (#2) was with the pK₅ adopting a non-anchor orientation exposed into the solvent. Both conformations (#1 and #2) had similar docking scores (with #1 only slightly higher), and both conformations would undoubtedly disrupt the structure of the complex at the pMHC-TCR interface (see Fig. Extended Data Fig. 7c,d), leading to the observed loss of T cell activation. Since the complex between Hsf2_WT and H2-D^b was quite unstable, and we could not isolate it in sufficient quantity for study, we can judge the obtained models based on circumstantial evidence only. Based on the existing crystallographic data, the W73 side chain conformation of H2-D^b is unaffected by the nature of a bound peptide (see an example in Fig. Extended Data Fig. 4c-e). This is consistent with a generally accepted notion that the MHC-peptide promiscuity is mainly driven by the plasticity of the peptides (<https://www.ncbi.nlm.nih.gov/pmc/articles/PMC3311345/>). From this perspective, the non-anchor conformation for pK₅ in the Hsf2_WT/H2-D^b complex seems more likely (Extended Data Fig. 6c-e) , but the question will remain unresolved until more data becomes available.

It is noteworthy that the docking between H2-D^b and the Hsf2 p.K72N peptide with pN₅ either in anchor or in the non-anchor position always resulted in a conformation similar to the crystal structure 7N9J with the pN₅ side chain in the anchor position and hydrogen-bonded to Q97. The docking score value for Hsf2 p.K72N was also significantly greater than that for Hsf2_WT (Extended Data Fig. 7b). In the course of the study the question was raised why replacement of pN₅ with certain bulky and aromatic side chains does not abolish TCR_47BE7-driven T cell activation. Again, molecular modeling suggested that such side chains can be accommodated in the H2-D^b pocket E upon rotation of the Y156 side chain towards

pocket D. One such example is shown in Extended Data Fig. 8d. We consider this conformation as likely, since Y156 can in fact adopt the alternate side chain conformations, as we have observed in the pMHC and the pMHC-TCR_47BE7 complex (Extended Data Fig. 5c,d), and as it was observed in the PDB structure 1BZ9 (Extended Data Fig. 8e), where the pF6 is the anchor residue and is situated inside the pocket E (<https://rupress.org/jem/article/189/2/359/7848/Structural-Evidence-of-T-Cell-Xeno-reactivity-in>).

- p 6: This page starts with a 12 line description of how peptides lie in the MHC groove. All very interesting, but presented as a hypothesis, later validated by structure determination. Being an experimentalist, I would normally determine the structure first and work out all this description later, with no need to hypothesise. However, I would accept that this was the genuine sequence of events, but I am just checking. If the structure determination did indeed precede the description, this para needs to be reworded please.

The reviewer raises a valid point. This was the genuine sequence of events. Based on known H2-D^b-specific epitope motifs (for example, <https://www.sciencedirect.com/science/article/pii/S0092867494901716>), we expected that the K72N mutation would result in the anchor pN₅, which in turn could lead to strong interactions between the Hsf2 neoepitope and H2-D^b. Subsequently, this prediction was validated by the X-ray crystallography in our study.

- p 6, 8 lines from the bottom: It is not at all surprising that the binding in pocket B is low affinity. pG2 does not have a side chain, therefore there is limited chance of interaction with the MHC. This pocket can take any side chain, and the general design of the MHC makes it imperative that the small side chains would leave cavities. But these cavities are solvent accessible, at least before folding. So solvent molecules may get trapped in there, but that is incidental. It also allows the MHC to be a general presenter of antigens.

We agree with the reviewers conclusions. However, the “loose” fitting of the epitope N-terminus inside the peptide binding pocket of H2-Db is clearly reflected by the increased thermal motion of the epitope and the adjacent H2-D^b residues (Fig. 5c) Moreover, the TCR does not interact with the N-terminus of the peptide-MHC complex, but binds to the peptide residues P₄-P₈, situated in the most rigid part of the pMHC-peptide interface. Based on the published data, binding along the least flexible interface may provide the stronger contact in the protein complex (<https://www.ncbi.nlm.nih.gov/pmc/articles/PMC4523921/>).

- p 6, last line: Residue 5 in the peptide is not normally an anchor residue, as I argue above. I leave it to the authors to consider rewording this statement. Indeed, on page 7, this residue is described a 'secondary anchor'!

Thank you for raising this excellent point. We have now clarified that, in the context of H2-D^b, residue p5 is serving as an anchor residue and have supported this previously observed concept with a citation (<https://translational-medicine.biomedcentral.com/articles/10.1186/s12967-021-02757-x>) in the text. This has also been observed in the following PDB entries: 5WLG, 7JWJ, 5M00, 7N4K, and 7NA5. We have also removed mention of it as a secondary anchor.

- p 7, l 3: 'W73-Y147-W147' should be 'W73-W147-Y156'

The text has been modified according to the reviewer's suggestion.

- p 7, l 7: 'DFW' should be 'DWF', as defined 2 lines before. The argument that follows seems to emphasise that the stability of the peptide binding into the groove is a good thing. That notion leaves me cold. The TCR response is no respecter of peptide-MHC stability, otherwise we would not observe allergies triggered by self-recognition after incidental cross-reactivity (diabetes). Also, in one case, a 'stabilised' peptide by changing pA2 to pL2, activated a different TCR which did not recognise the wt peptide as well (melanoma); the correct activation was dependent on extracting the N-terminus of the peptide out of the MHC pocket upon engagement by the cognate TCR. The authors may want to change the tone of this para to shift away from an argument on the peptide stabilisation. A good response to the mutant neoAg requires a DIFFERENT TCR, which must not engage the wt peptide. The authors have identified one, and I congratulate them.

We have changed notation throughout the manuscript to use B-factor instead of DWF. We respectfully disagree with the reviewer on this notion of stability. The peptide-MHC stability (which directly depends on affinity between peptide and MHC) is one of the main factors determining peptide immunogenicity (<https://www.ncbi.nlm.nih.gov/pmc/articles/PMC3872965/>). Reduced MHC-peptide complex stability would lead to the reduced epitope surface density and also may affect the stability of the TCR-MHC complex directly, as shown by Brian Baker (<https://www.pnas.org/doi/abs/10.1073/pnas.2018125118>). The following publications provide further robust evidence for the aforementioned notion of the importance of pMHC stability:

<https://pubmed.ncbi.nlm.nih.gov/12594952/>

<https://pubmed.ncbi.nlm.nih.gov/7809136/>

<https://www.ncbi.nlm.nih.gov/pmc/articles/PMC2193521/>

<https://pubmed.ncbi.nlm.nih.gov/14690592/>

We agree with the reviewer's assertion that a different TCR is likely necessary to react with the Hsf2₆₈₋₇₆ wild-type peptide.

- p 7, last para: The 'p6-p8 ridge' is not unique. It occurs in the human NYESO-1 peptide

recognised by 1G4, again propped up by a Trp side chain in the MHC. In that case, the one I am familiar with, residues 4 and 5 (MW) are pointing outwards.

We respectfully disagree. The p6-p8 ridge is typical for H2-D^b, but not for other MHCs. H2-D^b exhibits a (bulky) Trp73 as compared to Ser73 observed in H2-K^b and some human HLAs. The Trp73 together with Trp147 (the latter is very common in most MHC) props up the p6-p8 residues to form a “bulge”, which is often a main recognition pattern for TCR. There are at least 16 crystal structures for the pH2-D^b-TCR complexes (with 6 different peptides or its point mutants) (<https://tcr3d.ibbr.umd.edu/class1>), in which the p6-p8/p9 residues form a bulge that comprises the recognition pattern for each TCR.

- p 8, para 1, last line: 'confirmation' ==> 'conformation'
Confirmation was changed to conformation.

- p 8, para 2, l 2: 'G18-E19' should be 'E18-E19', as residue 18 is described as E 2 lines later. This is also its assignment in the deposited model.
Thank you for pointing this out. We have fixed this issue.

- p 8, para 2, l 4: 'vis-a-vis the backbone C-alpha towards while' does not make sense. This sentence needs to be recast.
We removed this sentence; further, the entire section has been changed significantly since the initial submission.

- p 10, para 2, last sentence: I disagree with this statement. Any side chain other than Asn at position 5 would not have the same conformation, because Asn makes specific contacts with the side chain of MHC residue Q97. The list of mutations tested cannot make that specific interaction. BLI may have detected recognition by TCR, but it cannot inform on peptide conformation. Size, shape and charge pattern are what matter.
We agree with the reviewer that a different side chain at P5 could have a different conformation from that of the N₅ side chain. However, if the mutated P5 residue retains its anchor conformation similar to that of pN₅, then its side chain could be accommodated inside the pocket C or E, whereas the backbone peptide conformation would remain essentially the same or very similar. This would result in the same shape and, subsequently, retain the TCR recognition pattern. The effect of the each substitution was measured by T cell activation (not by BLI), and these data indicated that among the P5 substitutions, only lysine and leucine completely abrogated T cell activation via TCR-47BE7 (Fig. 4f).

- p 10, last para: Again, hypothesis does not necessarily match the facts. pK5 being an outlier is probably a case in point, should that structure be determined.

We agree with the reviewer. The text has been revised accordingly. The modeling approach and the relevant data and revisions were discussed in detail above.

- p 11, para 1: At last, the authors recognise that pK5 does not fit in the pocket. The response, though, is surreal. They propose that the binding pocket needs to reorganise itself! They propose rotating the side chain of W73 by 180 deg, and cause wholesale rearrangement in the pocket, just to keep the K5 side chain tucked in. No, it's the peptide that would adopt a new conformation. The MHC fold is very stable, while the peptide is malleable. The chase after a conserved main chain for the peptide is like barking up the wrong tree. This conclusion is definitely wrong. It is not accidental that the pK5 variant is the outlier in the energy interaction calculations and measurements.

We agree with the reviewer that the conclusion regarding the mechanism of epitope recognition by H2-D^b cannot be based solely on the molecular modeling. This and the relevant issues have been addressed and are discussed above.

- Fig 1 legend: What is BA-Rank?

BA-rank is now defined in the Fig 1 legend as "Binding Affinity"-rank, which was a normalized binding affinity parameter output value produced by the NetMHCpan software.

- Fig 2, legend: Again, ACT not defined

Figure 2 legend is now edited such that ACT is defined as "Adoptive Cell Transfer", referring to Hsf2-reactive T cells that were transferred intravenously into tumor bearing mice.

- Fig 3, panel c: pN5 is shown contacting residue 'N97' which is actually 'Q97'

This has been corrected in the figure to show Q97 contacting pN5.

- Fig 3, panel f and its legend: The thermal parameters are referred to with the commonly used B-factor. The discussion in the main text on page 7 calls them DWF. The authors should use one style of reference to this quantity.

We have replaced all references to these parameters to universally refer to them as B-factors. All references to DWF have been removed.

Reviewer #2 (Remarks to the Author):

Finnigan et al. focused on the specificity and cross-reactivity of T cells towards neoantigens (neoAgs) and performed a wide range of experimental methods including RNA-seq, x-ray crystallography, in silico study, peptide library, and in vivo evaluations. The authors successfully identified neoAgs of B16F10, a common murine implantable tumor model of melanoma, using combined techniques. Furthermore, a high-affinity TCR targeting H2-Db-

restricted neoAg, Hsf2K72N, was identified. This TCR, TCR 47BE7, showed specific recognition of neoAg, Hsf2K72N of B16F10 in vitro and in vivo. Crystal structure of the peptide-MHC (pMHC, neoAg, Hsf2K72N/H2-Db) revealed that the K72N mutation makes new interaction with Asn at the bottom of the MHC groove (but not to TCR directly, categorized as class II), inducing C-terminal site of the peptide (pV6-pH8) rigid to form a solvent-exposed hydrophobic arch in H2-Db. Moreover, the crystal structure of the peptide-MHC-TCR complex (neoAg, Hsf2K72N/H2-Db, TCR 47BE7) complex showed that TCR 47BE7 mainly recognized this C-terminal site. In addition, this recognition is similar to that of an influenza peptide-H2-Db, suggesting common structural features between neoantigens and viral peptides. The study is extensive and interesting. I have some comments as follows.

1) The SPR analysis for TCR 47BE7 binding to Hsf2WT/H2-Db is helpful for understanding the functional difference between wild-type and K72N mutant quantitatively, even though Fig. 4g showed that WT peptide did not exhibit IFN γ production.

We thank the reviewer for their comments. We agree that such data could be useful, but we did not perform the binding studies with WT peptide. The reason is that the MHC complex with the WT peptide is not stable as a soluble protein and cannot be produced by conventional methods for SPR/BLI analysis.

2) Fig. 2b-d; only the pM order of the peptides can induce IFN γ production of neoantigen-reactive T cells, even though the KD of TCR-pMHC binding is μ M level. Did transgenic T cells significantly express TCRs?

This is a great point; to ensure robust expression of the transgenic TCR (tgTCR), we adopted a high-titer retroviral tgTCR transduction protocol, supplemented with a CRISPR:Cas9-mediated knockout of TRAC and TRBC to prevent potential endogenous TCR competition or mispairing (Extended Data Fig. 2a,b). We have added flow cytometric data to the supplemental figures indicating that all transgenic, engineered tumor-reactive T cells exhibited strong and equal TCR expression (Extended Data Fig. 2c). However, it is known in the literature that there is a discrepancy between the amount of peptide added to culture and what is ultimately presented on the cell surface. Others have reported that picomolar concentrations of peptide are required to generate efficient TCR responses (<https://www.sciencedirect.com/science/article/pii/S1074761300804471>).

3) Please add a figure showing a structural comparison of the C-terminal peptide region between this neoAg and an influenza peptide-H2-Db for readers to easily understand.

We have added the figures showing superposition between the H2-Db/Hsf2 p.K72N structure and the PDB structures with different peptides, including a separate figure for flu NP-N3D peptide with pN₅ (Extended Data Fig. 4c-e), PDB 48LC (<https://www.nature.com/articles/ncomms3663>). However, due to reviewers 1's

suggestion to focus less on this connection the focus on this observation is now only mentioned briefly in the manuscript.

4) Fig. 3c; The side chain of pN5 is red-based colored. However, oxygens are also colored red and thus a bit difficult to discriminate.

The color of pN₅ was changed to magenta.

5) Fig. 4c; Two CDR1βs exist.

We have corrected this issue and have revised the figure, showing both CDR1α and CDR1β (now Fig. 5d).

Reviewer #3 (Remarks to the Author):

The manuscript by Finnigan and colleagues examines the B16F10 murine tumor and isolates several candidate mutated peptides, also called neoantigens, that are expressed by the tumor. In particular, they focus on TCR 47BE7 which targets H2-Db-restricted neoAg Heat Shock Protein 2 (Hsf2 p.K72N68-76). With a series of biophysical measurements, the manuscript attempts to define the rules governing neoantigen immunogenicity with the anticipation of these rules being clinically relevant. *Overall, the paper is well written, but suffers in novelty and (large portions) being descriptive. Major concerns are as follows;*

We thank the reviewer for this feedback. The novelty of this story is the identification and evaluation of neoantigens in the B16F10 melanoma model. B16F10 is among the most widely used implantable tumor models in the scientific literature. This is significant, as previously the main models for tumor antigens in B16F10 consisted of either tumor-associated antigens (gp100/pmel, for example) or artificial foreign antigen models (B16F10-OVA, for example). We present to the community a wide range of neoantigens and respective neoantigen-reactive TCRs/CD8+ T cells that vary in their cross-reactivity towards WT sequences, pMHC-TCR affinities and avidities, MHC restriction, etc. Of particular interest, we identified a particularly high affinity TCR, 47BE7, that recognizes the H2-D^b-restricted neopeptide YGFRNVVHI from the protein Hsf2. Vaccination and ACT (under the conditions of high neoantigen expression) exhibited efficacy in mice. We present this model to the field to investigate how to improve and harness neoantigen-reactive immunity across various immunotherapeutic modalities, in a notoriously challenging (B16F10) model that mimics many features of human cancers (poor response to checkpoint blockade, fast progression, low immune infiltrate, etc.). Please see the edited introduction and discussion in particular, in which we discuss the points made above.

1) The predominant T cell (TCR) clone chosen for modeling, 47BE7, does not recognize the native tumor, unless the antigen is over-expressed. This is not addressed and it severely

impacts the rationale, especially since the peptide vaccination by itself is able to elicit some tumor control.

This is a great point, we have edited the text to highlight this condition for ACT efficacy more clearly, and have expanded upon this point in the discussion (see page 16). See response to question 4) below for our full response. [Combined response to address similar questions]. However, it should also be noted that in our study, TCR 47BE7 recognizes the native tumor *in vitro*, in a H2-D^b-dependent fashion (Fig 3a). Cytokine production triggered by the native tumor exceeds that observed using TCR 55BA5, which is specific to gp100/pmel, a common antigenic target in this model. Of note, in the B16F10, expression of Hsf2 (FPKM 6.28), is 246-fold lower than gp100/pmel (FPKM 1551.16) (Extended Fig 2d).

2) Figure 3 and 4 are largely descriptive and makes some predictions and correlations. However, none of these are truly tested and so the rules defining neoantigens, which the manuscript sets out to define, are left hanging in the balance.

We agree with the reviewer. All the predictions and correlation data were moved into the Extended Data section of the manuscript (Extended Data Figs. 4, 6, 7, 8). The main figures were reformatted to include only the experimental results. We also have reframed the manuscript such that we are not defining neoantigen “rules”.

3) Figure 2h,i needs ACT of tumor-nonspecific T cells as a control not vehicle. This experiments should have been repeated with all the candidate peptides/TCR identified in Fig2a. Tumor rejection is a far more important parameter clinically than simply measuring IFNg

We have added the data showing that control, non-tumor specific, OVA-reactive T cells (which recognize the SIINFEKL peptide from the protein ovalbumin) do not reduce tumor growth of Hsf2-overexpressing (OE) B16F10, while Hsf2-reactive T cell ACT does (now Fig. 3d,e). We refrained from repeating ACT experiments with all candidate peptide/TCRs because apart from Hsf2, candidate TCRs did not recognize B16F10 *in vitro* (i.e., complete lack of IFNg upregulation; data shown in Fig. 3a), precluding any type of T cell-specific recognition and subsequent tumor cell killing *in vitro* or *in vivo*.

4) It is difficult to understand the clinical relevance of this study given that the TCR that is the primary focus of the study does not reject the parental tumor.

This is a very important point and we have modified our writing to better convey our rationale in conducting this study and limitations of this study. Our study was not designed towards immediate, direct translatable clinical applications, but rather to be a model for studying features of potent neoantigen and neoantigen-reactive TCRs in a notoriously poorly immunogenic *in vivo* melanoma model, B16F10. While ACT response was dependent upon artificially-induced high expression of Hsf2_{K72N}, vaccination elicited modest, but significant, tumor growth control in WT B16F10. We

view our ACT data as a proof-of-concept experiment that, given the right conditions, *in vivo* recognition and killing of B16F10 tumor by Hsf2-reactive T cells can occur. This data we view as additional evidence (i.e., aside from vaccination) that Hsf2-reactive T cells can recognize and kill tumor cells in the context of various classes of immunotherapeutic modalities. In particular, ACT is known to have unique challenges; many existing models exhibit poor tumor growth control unless modifications or combinatorial strategies are employed (<https://insight.jci.org/articles/view/124405/figure/2>). Commonly studied ACT models for the melanoma antigens MART-1 and gp100 and their respective tumor-reactive T cells require modification and/or overexpression for ACT efficacy (<https://www.ncbi.nlm.nih.gov/pmc/articles/PMC2267026/>). Further, in a gp100-reactive T cell model in which gp100 is artificially overexpressed, tumor growth reduction, but not regression, is observed, similar to that observed in our model, and is only further improved upon addition of a BRAF inhibitor (<https://www.ncbi.nlm.nih.gov/pmc/articles/PMC4120472/>), demonstrating that even under optimal tumor antigen expression conditions, ACT can be very difficult to render maximally effective. Future studies are underway in our laboratory to address how to improve ACT efficacy in the context of low neoantigen density. This problem has been acknowledged by others (<https://www.ncbi.nlm.nih.gov/pmc/articles/PMC8562866/>), and we will use this novel model to address this issue, particularly in the ACT context. Vaccination and ACT we view as apples vs. oranges; they exhibit differing kinetic courses, and different requirements for therapeutic success. Therefore, for the purposes of this study, we believe our findings of the structural properties of a high avidity neoantigen-reactive TCR-pMHC complex are relevant for the field, irrespective of ACT efficacy against wild type tumors. Our structural work opens up the possibility for further innovative engineering strategies to provide solutions for overcoming current limitations of ACT, that include functional avidity and modification of T cell potency.

5) How universal the putative neoantigen-defining rules are to the next TCR and the next tumor neoantigen/MHC is unclear, or will these putative rules be specific to this one p/MHC/TCR?

We have reworded our discussion; instead of creating “neoantigen-defining rules”, we instead discuss our findings in the context of previous work in the field and clinical relevance. The antigen-TCR recognition pattern defined herein is not unusual for H2-D^b-restricted peptides. However, each H2-D^b pMHC-TCR structure has unique features, and to this day, a limited number of such structures has been published. In this case the main feature is not the structure itself, but the data showing that the WT peptide does not bind efficiently to H2-D^b, and when it binds, it is modeled to form a complex topologically distinct from the mutant pN₅ neoepitope. Our data bolsters the

notion that group II (anchor-residue modified) neoantigens can be surgically recognized by its cognate TCR, without significant cross-reactivity to WT epitopes (<https://www.frontiersin.org/articles/10.3389/fimmu.2022.833017/full>). Thus, these antigens may be efficient targets for cancer immunotherapy. In sum, the significance of our findings is not limited to this TCR; the structural nuances in recognition of H2-D^b/Hsf2 p.K72N should be noted for future studies, while at the same, we have added to the body of evidence that group II anchor residue-modified neoantigens can be good targets for T cell-mediated cancer immunotherapy.

Reviewer #4 (Remarks to the Author):

The authors provide a comprehensive analysis of neoantigen recognition in the murine B16F10 melanoma model, starting with exome sequencing to define epitopes and subsequent T cell cloning and TCR characterization. This is an impressive compilation of functional and molecular immunology that provides insight into tumor neoantigen recognition. The biophysics and structural biology is thorough and of high quality.

There is a significant amount of data generated and presented in this analysis, so understandably there are challenges in data presentation. However, the figures are currently very difficult to read and will be even more difficult to read when formatted for a publication. Perhaps moving some of the data to the supplement might help. The font size should also be increased so that the reader does not have to zoom in on a pdf, as this will make reading on a printout almost impossible. Other than this, the manuscript provides an abundance of interesting molecular and biophysical data that will add to our understanding of neoantigen recognition in the anti-tumor T cell mediated response.

We thank the reviewer for their positive feedback. We have increased figure and font sizes throughout the paper, and have divided data into additional main figures. Further, we have rearranged our supplemental data in part in response to the reviewer's comments as well; in particular, we have moved all modeling data to the supplement, while keeping experimental structural data in the main figure section.

REVIEWER COMMENTS

Reviewer #1 (Remarks to the Author):

The authors have answered my comments appropriately. Amendments were made in the light of my remarks, improving the manuscript in the process. The manuscript appears to be sound, with reasonable arguments and conclusions. I still disagree with some points, but they fall into the category of difference in opinion and do not impact the worth of the submission. My recommendation is for acceptance after the revisions that have been made.

REVIEWER COMMENTS & RESPONSES

We thank the reviewers for their comments and address these below in response to each reviewer's critiques.

We have made extensive changes throughout the manuscript to improve it in response to reviewers' comments, including making figures larger and more legible. We completely reworked the structural section of the manuscript, adding extensive modeling analyses based on published datasets and moving this data to the supplemental section, as suggested. We also edited our structural figures in the main text for better clarity. Our discussion section of the manuscript has been extensively modified and extended to fully capture the novelty of the study, data interpretation in the broader context of other published studies, and future directions. We believe these changes have vastly improved the quality of our manuscript and thank the reviewers for their thoughtful and helpful comments.

Reviewer #1 (Remarks to the Author):

This manuscript discusses a well conducted experiment that has paid attention to detail. The quality of the results are excellent. Some computer aided modelling was used to extend the results and make predictions or extrapolations that are well informed, but the conclusions drawn are not safe enough, in my opinion. These will need some attention, and perhaps modification in the light of my comments below. There are also a few editorial corrections required. My recommendation is that the manuscript should be accepted for publication, if the editor is satisfied that my suggestions for changes are met.

Notes for the authors - Please note that in the pdf document I got, the main text starts at page 3.

=====

We thank reviewer #1 for their thoughtful comments and recognition of the quality of results. We respond below to each critique and believe we have addressed each issue.

- Abstract: shared viral immunogenicity is topical, but might be a distraction. The main text only skims this like an afterthought, and does not have a serious discussion around this point.

We agree and have removed this concept from the abstract and significant conjectures from the discussion as well. We recognize that the shared motif is a feature that has been observed for H2-D^b-peptide complexes and that this does not necessarily signify any likeness between influenza and melanoma antigens. Separate from this however is the notion that Hsf2 p.K72N behaves as a "non-self" peptide because it is selectively recognized, with little cross-reactivity to the wild type peptide, by its cognate TCR, 47BE7.

- Introduction, p 4: Would the authors elaborate on the reason for restricting the investigation to 7 neoAg?

We elaborated upon the selection of neoAg in the introduction. After computational prediction of neoantigen pMHC complexes, we vaccinated with 12 predicted long peptide neoAgs. H2-K^b or H2-D^b restriction and specific minimal peptide epitope determination was resolved using a panel of 50+ MHC tetramers; from this, we observed only seven MHC-I restricted pMHCs elicited T cell responses, hence the decision to restrict the following studies to 7 neoantigens (Fig. 1c,d and Extended Data Fig. 1).

- p 5, para 2, 'used to organ neoAg into two principal classes': Presumably 'organ' should be 'organise'. I prefer the use of 'groups' instead of 'classes' to avoid confusion with Class I/II MHC. This is very relevant at the bottom of p5: 'prototype murine class II neoAg' and further, particularly in the discussion section towards the end of the text.

All requested changes have been addressed in the revised text; "class" has now been replaced by "group" where relevant.

5. - p 5, last para: The discussion comparing pK5 and pN5 is shielding the reader from an important point, relying on the credibility of in-silico modelling. I have no problem with that at all. In this case though, I have serious doubt that the conformation of the peptide would be the same when residue 5 is the wt type K instead of N as in this study. I assume that the in-silico modelling simply replaced the side chain of Asn as determined from the experiment, by coordinates for a Lys, before running a molecular dynamics minimisation. This indeed would show Asn to be much more favourable, as its side chain perfectly complements the side chain of Gln97, both in size and charge complementarity, a perfect fit. But altering the side chain to be a Lys immediately creates a clash with Gln97, because its side chain is too long. It has to bend away from this side chain. Being a charged residue, it can only move towards Glu9, a little too far, or Gln70, still not a perfect fit. In my opinion, this side chain would instead point outwards from the groove, like pR4. Many other peptides are presented thus, with all central residues pointing outwards, what I describe as the OMEGA peptide conformation, as opposed to the M conformation as in this case. This would make perfect sense, and would also explain the failure of the wt peptide to activate the identified TCR. If anything, this could be a perfect demonstration of the basis for the thymus to eliminate any TCR that would recognise the wt peptide (self) while permitting TCRs that recognise the mutant. This conjecture can be settled by determining the structure of the MHC presenting the wt peptide. If that structure had been determined, it is not being discussed in this manuscript. As such, the experiment is incomplete, making the conclusions a little unsafe. It may be an oversight, or the circumstances may have militated against determining that extra structure, and I would not insist on having it completed for the sake of the paper, but the authors may wish to re-examine their comments pursuing my notes.

We thank the reviewer for these thoughtful comments and agree that while molecular modeling predicts possible conformations, it cannot be used to make conclusions. The modeling data have to be corroborated with the existing experimental data offering at least indirect support of the models. First, we performed a more detailed analysis of modeling data produced by the PepFlexDock server. Using the coordinates of the solved crystal structure between H2-D^b and Hsf2 p.K72N, we produced a series of models in which pN₅ of the epitope is replaced with other amino acid residues, including lysine. For each model two structural peptide variants were designed, in which the residue at P5 adopted either anchor (1) or non-anchor (2) orientations, respectively. The vast majority of the substitutions resulted in the amino acid residue at P5 adopting anchor conformations in the highest scoring solutions. Such modeling predictions were in a complete agreement with the available crystallographic data (see an example presented in Extended Data Fig. 6c-e). By contrast, there were two types of the top scored conformations generated between Hsf2-WT peptide (pK₅) and H2-D^b – both with similar docking scores. In model #1 the pK₅ side chain remained in the anchor position, but the H2-D^b W73 was displaced from the pocket E to accommodate the lysine side chain. This conformation was shown in the previous version of the manuscript. However, the alternate Hsf2_WT conformation (#2) was with the pK₅ adopting a non-anchor orientation exposed into the solvent. Both conformations (#1 and #2) had similar docking scores (with #1 only slightly higher), and both conformations would undoubtedly disrupt the structure of the complex at the pMHC-TCR interface (see Fig. Extended Data Fig. 7c,d), leading to the observed loss of T cell activation. Since the complex between Hsf2_WT and H2-D^b was quite unstable, and we could not isolate it in sufficient quantity for study, we can judge the obtained models based on circumstantial evidence only. Based on the existing crystallographic data, the W73 side chain conformation of H2-D^b is unaffected by the nature of a bound peptide (see an example in Fig. Extended Data Fig. 4c-e). This is consistent with a generally accepted notion that the MHC-peptide promiscuity is mainly driven by the plasticity of the peptides (<https://www.ncbi.nlm.nih.gov/pmc/articles/PMC3311345/>). From this perspective, the non-anchor conformation for pK₅ in the Hsf2_WT/H2-D^b complex seems more likely (Extended Data Fig. 6c-e), but the question will remain unresolved until more data becomes available.

It is noteworthy that the docking between H2-D^b and the Hsf2 p.K72N peptide with pN₅ either in anchor or in the non-anchor position always resulted in a conformation similar to the crystal structure 7N9J with the pN₅ side chain in the anchor position and hydrogen-bonded to Q97. The docking score value for Hsf2 p.K72N was also significantly greater than that for Hsf2_WT (Extended Data Fig. 7b). In the course of the study the question was raised why replacement of pN₅ with certain bulky and aromatic side chains does not abolish TCR_47BE7-driven T cell activation. Again, molecular modeling suggested that such side chains can be accommodated in the H2-D^b pocket E upon rotation of the Y156 side chain towards

pocket D. One such example is shown in Extended Data Fig. 8d. We consider this conformation as likely, since Y156 can in fact adopt the alternate side chain conformations, as we have observed in the pMHC and the pMHC-TCR_47BE7 complex (Extended Data Fig. 5c,d), and as it was observed in the PDB structure 1BZ9 (Extended Data Fig. 8e), where the pF6 is the anchor residue and is situated inside the pocket E (<https://rupress.org/jem/article/189/2/359/7848/Structural-Evidence-of-T-Cell-Xeno-reactivity-in>).

- p 6: This page starts with a 12 line description of how peptides lie in the MHC groove. All very interesting, but presented as a hypothesis, later validated by structure determination. Being an experimentalist, I would normally determine the structure first and work out all this description later, with no need to hypothesise. However, I would accept that this was the genuine sequence of events, but I am just checking. If the structure determination did indeed precede the description, this para needs to be reworded please.

The reviewer raises a valid point. This was the genuine sequence of events. Based on known H2-D^b-specific epitope motifs (for example, <https://www.sciencedirect.com/science/article/pii/0092867494901716>), we expected that the K72N mutation would result in the anchor pN₅, which in turn could lead to strong interactions between the Hsf2 neoepitope and H2-D^b. Subsequently, this prediction was validated by the X-ray crystallography in our study.

- p 6, 8 lines from the bottom: It is not at all surprising that the binding in pocket B is low affinity. pG2 does not have a side chain, therefore there is limited chance of interaction with the MHC. This pocket can take any side chain, and the general design of the MHC makes it imperative that the small side chains would leave cavities. But these cavities are solvent accessible, at least before folding. So solvent molecules may get trapped in there, but that is incidental. It also allows the MHC to be a general presenter of antigens.

We agree with the reviewers conclusions. However, the “loose” fitting of the epitope N-terminus inside the peptide binding pocket of H2-Db is clearly reflected by the increased thermal motion of the epitope and the adjacent H2-D^b residues (Fig. 5c) Moreover, the TCR does not interact with the N-terminus of the peptide-MHC complex, but binds to the peptide residues P₄-P₈, situated in the most rigid part of the pMHC-peptide interface. Based on the published data, binding along the least flexible interface may provide the stronger contact in the protein complex (<https://www.ncbi.nlm.nih.gov/pmc/articles/PMC4523921/>).

- p 6, last line: Residue 5 in the peptide is not normally an anchor residue, as I argue above. I leave it to the authors to consider rewording this statement. Indeed, on page 7, this residue is described a 'secondary anchor'!

Thank you for raising this excellent point. We have now clarified that, in the context of H2-D^b, residue p5 is serving as an anchor residue and have supported this previously observed concept with a citation (<https://translational-medicine.biomedcentral.com/articles/10.1186/s12967-021-02757-x>) in the text. This has also been observed in the following PDB entries: 5WLG, 7JWJ, 5M00, 7N4K, and 7NA5. We have also removed mention of it as a secondary anchor.

- p 7, l 3: 'W73-Y147-W147' should be 'W73-W147-Y156'

The text has been modified according to the reviewer's suggestion.

- p 7, l 7: 'DFW' should be 'DWF', as defined 2 lines before. The argument that follows seems to emphasise that the stability of the peptide binding into the groove is a good thing. That notion leaves me cold. The TCR response is no respecter of peptide-MHC stability, otherwise we would not observe allergies triggered by self-recognition after incidental cross-reactivity (diabetes). Also, in one case, a 'stabilised' peptide by changing pA2 to pL2, activated a different TCR which did not recognise the wt peptide as well (melanoma); the correct activation was dependent on extracting the N-terminus of the peptide out of the MHC pocket upon engagement by the cognate TCR. The authors may want to change the tone of this para to shift away from an argument on the peptide stabilisation. A good response to the mutant neoAg requires a DIFFERENT TCR, which must not engage the wt peptide. The authors have identified one, and I congratulate them.

We have changed notation throughout the manuscript to use B-factor instead of DWF. We respectfully disagree with the reviewer on this notion of stability. The peptide-MHC stability (which directly depends on affinity between peptide and MHC) is one of the main factors determining peptide immunogenicity (<https://www.ncbi.nlm.nih.gov/pmc/articles/PMC3872965/>). Reduced MHC-peptide complex stability would lead to the reduced epitope surface density and also may affect the stability of the TCR-MHC complex directly, as shown by Brian Baker (<https://www.pnas.org/doi/abs/10.1073/pnas.2018125118>). The following publications provide further robust evidence for the aforementioned notion of the importance of pMHC stability:

<https://pubmed.ncbi.nlm.nih.gov/12594952/>

<https://pubmed.ncbi.nlm.nih.gov/7809136/>

<https://www.ncbi.nlm.nih.gov/pmc/articles/PMC2193521/>

<https://pubmed.ncbi.nlm.nih.gov/14690592/>

We agree with the reviewer's assertion that a different TCR is likely necessary to react with the Hsf2₆₈₋₇₆ wild-type peptide.

- p 7, last para: The 'p6-p8 ridge' is not unique. It occurs in the human NYESO-1 peptide

recognised by 1G4, again propped up by a Trp side chain in the MHC. In that case, the one I am familiar with, residues 4 and 5 (MW) are pointing outwards.

We respectfully disagree. The p6-p8 ridge is typical for H2-D^b, but not for other MHCs. H2-D^b exhibits a (bulky) Trp73 as compared to Ser73 observed in H2-K^b and some human HLAs. The Trp73 together with Trp147 (the latter is very common in most MHC) props up the p6-p8 residues to form a “bulge”, which is often a main recognition pattern for TCR. There are at least 16 crystal structures for the pH2-D^b-TCR complexes (with 6 different peptides or its point mutants) (<https://tcr3d.ibbr.umd.edu/class1>), in which the p6-p8/p9 residues form a bulge that comprises the recognition pattern for each TCR.

- p 8, para 1, last line: 'confirmation' ==> 'conformation'
Confirmation was changed to conformation.

- p 8, para 2, l 2: 'G18-E19' should be 'E18-E19', as residue 18 is described as E 2 lines later. This is also its assignment in the deposited model.
Thank you for pointing this out. We have fixed this issue.

- p 8, para 2, l 4: 'vis-a-vis the backbone C-alpha towards while' does not make sense. This sentence needs to be recast.
We removed this sentence; further, the entire section has been changed significantly since the initial submission.

- p 10, para 2, last sentence: I disagree with this statement. Any side chain other than Asn at position 5 would not have the same conformation, because Asn makes specific contacts with the side chain of MHC residue Q97. The list of mutations tested cannot make that specific interaction. BLI may have detected recognition by TCR, but it cannot inform on peptide conformation. Size, shape and charge pattern are what matter.
We agree with the reviewer that a different side chain at P5 could have a different conformation from that of the N₅ side chain. However, if the mutated P5 residue retains its anchor conformation similar to that of pN₅, then its side chain could be accommodated inside the pocket C or E, whereas the backbone peptide conformation would remain essentially the same or very similar. This would result in the same shape and, subsequently, retain the TCR recognition pattern. The effect of the each substitution was measured by T cell activation (not by BLI), and these data indicated that among the P5 substitutions, only lysine and leucine completely abrogated T cell activation via TCR-47BE7 (Fig. 4f).

- p 10, last para: Again, hypothesis does not necessarily match the facts. pK5 being an outlier is probably a case in point, should that structure be determined.

We agree with the reviewer. The text has been revised accordingly. The modeling approach and the relevant data and revisions were discussed in detail above.

- p 11, para 1: At last, the authors recognise that pK5 does not fit in the pocket. The response, though, is surreal. They propose that the binding pocket needs to reorganise itself! They propose rotating the side chain of W73 by 180 deg, and cause wholesale rearrangement in the pocket, just to keep the K5 side chain tucked in. No, it's the peptide that would adopt a new conformation. The MHC fold is very stable, while the peptide is malleable. The chase after a conserved main chain for the peptide is like barking up the wrong tree. This conclusion is definitely wrong. It is not accidental that the pK5 variant is the outlier in the energy interaction calculations and measurements.

We agree with the reviewer that the conclusion regarding the mechanism of epitope recognition by H2-D^b cannot be based solely on the molecular modeling. This and the relevant issues have been addressed and are discussed above.

- Fig 1 legend: What is BA-Rank?

BA-rank is now defined in the Fig 1 legend as "Binding Affinity"-rank, which was a normalized binding affinity parameter output value produced by the NetMHCpan software.

- Fig 2, legend: Again, ACT not defined

Figure 2 legend is now edited such that ACT is defined as "Adoptive Cell Transfer", referring to Hsf2-reactive T cells that were transferred intravenously into tumor bearing mice.

- Fig 3, panel c: pN5 is shown contacting residue 'N97' which is actually 'Q97'

This has been corrected in the figure to show Q97 contacting pN5.

- Fig 3, panel f and its legend: The thermal parameters are referred to with the commonly used B-factor. The discussion in the main text on page 7 calls them DWF. The authors should use one style of reference to this quantity.

We have replaced all references to these parameters to universally refer to them as B-factors. All references to DWF have been removed.

Reviewer #2 (Remarks to the Author):

Finnigan et al. focused on the specificity and cross-reactivity of T cells towards neoantigens (neoAgs) and performed a wide range of experimental methods including RNA-seq, x-ray crystallography, in silico study, peptide library, and in vivo evaluations. The authors successfully identified neoAgs of B16F10, a common murine implantable tumor model of melanoma, using combined techniques. Furthermore, a high-affinity TCR targeting H2-Db-

restricted neoAg, Hsf2K72N, was identified. This TCR, TCR 47BE7, showed specific recognition of neoAg, Hsf2K72N of B16F10 in vitro and in vivo. Crystal structure of the peptide-MHC (pMHC, neoAg, Hsf2K72N/H2-Db) revealed that the K72N mutation makes new interaction with Asn at the bottom of the MHC groove (but not to TCR directly, categorized as class II), inducing C-terminal site of the peptide (pV6-pH8) rigid to form a solvent-exposed hydrophobic arch in H2-Db. Moreover, the crystal structure of the peptide-MHC-TCR complex (neoAg, Hsf2K72N/H2-Db, TCR 47BE7) complex showed that TCR 47BE7 mainly recognized this C-terminal site. In addition, this recognition is similar to that of an influenza peptide-H2-Db, suggesting common structural features between neoantigens and viral peptides. The study is extensive and interesting. I have some comments as follows.

1) The SPR analysis for TCR 47BE7 binding to Hsf2WT/H2-Db is helpful for understanding the functional difference between wild-type and K72N mutant quantitatively, even though Fig. 4g showed that WT peptide did not exhibit IFN γ production.

We thank the reviewer for their comments. We agree that such data could be useful, but we did not perform the binding studies with WT peptide. The reason is that the MHC complex with the WT peptide is not stable as a soluble protein and cannot be produced by conventional methods for SPR/BLI analysis.

2) Fig. 2b-d; only the pM order of the peptides can induce IFN γ production of neoantigen-reactive T cells, even though the KD of TCR-pMHC binding is μ M level. Did transgenic T cells significantly express TCRs?

This is a great point; to ensure robust expression of the transgenic TCR (tgTCR), we adopted a high-titer retroviral tgTCR transduction protocol, supplemented with a CRISPR:Cas9-mediated knockout of TRAC and TRBC to prevent potential endogenous TCR competition or mispairing (Extended Data Fig. 2a,b). We have added flow cytometric data to the supplemental figures indicating that all transgenic, engineered tumor-reactive T cells exhibited strong and equal TCR expression (Extended Data Fig. 2c). However, it is known in the literature that there is a discrepancy between the amount of peptide added to culture and what is ultimately presented on the cell surface. Others have reported that picomolar concentrations of peptide are required to generate efficient TCR responses (<https://www.sciencedirect.com/science/article/pii/S1074761300804471>).

3) Please add a figure showing a structural comparison of the C-terminal peptide region between this neoAg and an influenza peptide-H2-Db for readers to easily understand.

We have added the figures showing superposition between the H2-Db/Hsf2 p.K72N structure and the PDB structures with different peptides, including a separate figure for flu NP-N3D peptide with pN₅ (Extended Data Fig. 4c-e), PDB 48LC (<https://www.nature.com/articles/ncomms3663>). However, due to reviewers 1's

suggestion to focus less on this connection the focus on this observation is now only mentioned briefly in the manuscript.

4) Fig. 3c; The side chain of pN5 is red-based colored. However, oxygens are also colored red and thus a bit difficult to discriminate.

The color of pN₅ was changed to magenta.

5) Fig. 4c; Two CDR1βs exist.

We have corrected this issue and have revised the figure, showing both CDR1α and CDR1β (now Fig. 5d).

Reviewer #3 (Remarks to the Author):

The manuscript by Finnigan and colleagues examines the B16F10 murine tumor and isolates several candidate mutated peptides, also called neoantigens, that are expressed by the tumor. In particular, they focus on TCR 47BE7 which targets H2-Db-restricted neoAg Heat Shock Protein 2 (Hsf2 p.K72N68-76). With a series of biophysical measurements, the manuscript attempts to define the rules governing neoantigen immunogenicity with the anticipation of these rules being clinically relevant. *Overall, the paper is well written, but suffers in novelty and (large portions) being descriptive. Major concerns are as follows;*

We thank the reviewer for this feedback. The novelty of this story is the identification and evaluation of neoantigens in the B16F10 melanoma model. B16F10 is among the most widely used implantable tumor models in the scientific literature. This is significant, as previously the main models for tumor antigens in B16F10 consisted of either tumor-associated antigens (gp100/pmel, for example) or artificial foreign antigen models (B16F10-OVA, for example). We present to the community a wide range of neoantigens and respective neoantigen-reactive TCRs/CD8+ T cells that vary in their cross-reactivity towards WT sequences, pMHC-TCR affinities and avidities, MHC restriction, etc. Of particular interest, we identified a particularly high affinity TCR, 47BE7, that recognizes the H2-D^b-restricted neopeptide YGFRNVVHI from the protein Hsf2. Vaccination and ACT (under the conditions of high neoantigen expression) exhibited efficacy in mice. We present this model to the field to investigate how to improve and harness neoantigen-reactive immunity across various immunotherapeutic modalities, in a notoriously challenging (B16F10) model that mimics many features of human cancers (poor response to checkpoint blockade, fast progression, low immune infiltrate, etc.). Please see the edited introduction and discussion in particular, in which we discuss the points made above.

1) The predominant T cell (TCR) clone chosen for modeling, 47BE7, does not recognize the native tumor, unless the antigen is over-expressed. This is not addressed and it severely

impacts the rationale, especially since the peptide vaccination by itself is able to elicit some tumor control.

This is a great point, we have edited the text to highlight this condition for ACT efficacy more clearly, and have expanded upon this point in the discussion (see page 16). See response to question 4) below for our full response. [Combined response to address similar questions]. However, it should also be noted that in our study, TCR 47BE7 recognizes the native tumor *in vitro*, in a H2-D^b-dependent fashion (Fig 3a). Cytokine production triggered by the native tumor exceeds that observed using TCR 55BA5, which is specific to gp100/pmel, a common antigenic target in this model. Of note, in the B16F10, expression of Hsf2 (FPKM 6.28), is 246-fold lower than gp100/pmel (FPKM 1551.16) (Extended Fig 2d).

2) Figure 3 and 4 are largely descriptive and makes some predictions and correlations. However, none of these are truly tested and so the rules defining neoantigens, which the manuscript sets out to define, are left hanging in the balance.

We agree with the reviewer. All the predictions and correlation data were moved into the Extended Data section of the manuscript (Extended Data Figs. 4, 6, 7, 8). The main figures were reformatted to include only the experimental results. We also have reframed the manuscript such that we are not defining neoantigen “rules”.

3) Figure 2h,i needs ACT of tumor-nonspecific T cells as a control not vehicle. This experiments should have been repeated with all the candidate peptides/TCR identified in Fig2a. Tumor rejection is a far more important parameter clinically than simply measuring IFN γ

We have added the data showing that control, non-tumor specific, OVA-reactive T cells (which recognize the SIINFEKL peptide from the protein ovalbumin) do not reduce tumor growth of Hsf2-overexpressing (OE) B16F10, while Hsf2-reactive T cell ACT does (now Fig. 3d,e). We refrained from repeating ACT experiments with all candidate peptide/TCRs because apart from Hsf2, candidate TCRs did not recognize B16F10 *in vitro* (i.e., complete lack of IFN γ upregulation; data shown in Fig. 3a), precluding any type of T cell-specific recognition and subsequent tumor cell killing *in vitro* or *in vivo*.

4) It is difficult to understand the clinical relevance of this study given that the TCR that is the primary focus of the study does not reject the parental tumor.

This is a very important point and we have modified our writing to better convey our rationale in conducting this study and limitations of this study. Our study was not designed towards immediate, direct translatable clinical applications, but rather to be a model for studying features of potent neoantigen and neoantigen-reactive TCRs in a notoriously poorly immunogenic *in vivo* melanoma model, B16F10. While ACT response was dependent upon artificially-induced high expression of Hsf2_{K72N}, vaccination elicited modest, but significant, tumor growth control in WT B16F10. We

view our ACT data as a **proof-of-concept** experiment that, given the right conditions, *in vivo* recognition and killing of B16F10 tumor by Hsf2-reactive T cells can occur. This data we view as additional evidence (i.e., aside from vaccination) that Hsf2-reactive T cells can recognize and kill tumor cells in the context of various classes of immunotherapeutic modalities. In particular, ACT is known to have unique challenges; many existing models exhibit poor tumor growth control unless modifications or combinatorial strategies are employed (<https://insight.jci.org/articles/view/124405/figure/2>). Commonly studied ACT models for the melanoma antigens MART-1 and gp100 and their respective tumor-reactive T cells require modification and/or overexpression for ACT efficacy (<https://www.ncbi.nlm.nih.gov/pmc/articles/PMC2267026/>). Further, in a gp100-reactive T cell model in which gp100 is artificially overexpressed, tumor growth reduction, but not regression, is observed, similar to that observed in our model, and is only further improved upon addition of a BRAF inhibitor (<https://www.ncbi.nlm.nih.gov/pmc/articles/PMC4120472/>), demonstrating that even under optimal tumor antigen expression conditions, ACT can be very difficult to render maximally effective. Future studies are underway in our laboratory to address how to improve ACT efficacy in the context of low neoantigen density. This problem has been acknowledged by others (<https://www.ncbi.nlm.nih.gov/pmc/articles/PMC8562866/>), and we will use this novel model to address this issue, particularly in the ACT context. Vaccination and ACT we view as apples vs. oranges; they exhibit differing kinetic courses, and different requirements for therapeutic success. Therefore, for the purposes of this study, we believe our findings of the structural properties of a high avidity neoantigen-reactive TCR-pMHC complex are relevant for the field, irrespective of ACT efficacy against wild type tumors. Our structural work opens up the possibility for further innovative engineering strategies to provide solutions for overcoming current limitations of ACT, that include functional avidity and modification of T cell potency.

5) How universal the putative neoantigen-defining rules are to the next TCR and the next tumor neoantigen/MHC is unclear, or will these putative rules be specific to this one p/MHC/TCR?

We have reworded our discussion; instead of creating “neoantigen-defining rules”, we instead discuss our findings in the context of previous work in the field and clinical relevance. The antigen-TCR recognition pattern defined herein is not unusual for H2-D^b-restricted peptides. However, each H2-D^b pMHC-TCR structure has unique features, and to this day, a limited number of such structures has been published. In this case the main feature is not the structure itself, but the data showing that the WT peptide does not bind efficiently to H2-D^b, and when it binds, it is modeled to form a complex topologically distinct from the mutant pN₅ neoepitope. Our data bolsters the

notion that group II (anchor-residue modified) neoantigens can be surgically recognized by its cognate TCR, without significant cross-reactivity to WT epitopes (<https://www.frontiersin.org/articles/10.3389/fimmu.2022.833017/full>). Thus, these antigens may be efficient targets for cancer immunotherapy. In sum, the significance of our findings is not limited to this TCR; the structural nuances in recognition of H2-D^b/Hsf2 p.K72N should be noted for future studies, while at the same, we have added to the body of evidence that group II anchor residue-modified neoantigens can be good targets for T cell-mediated cancer immunotherapy.

Reviewer #4 (Remarks to the Author):

The authors provide a comprehensive analysis of neoantigen recognition in the murine B16F10 melanoma model, starting with exome sequencing to define epitopes and subsequent T cell cloning and TCR characterization. This is an impressive compilation of functional and molecular immunology that provides insight into tumor neoantigen recognition. The biophysics and structural biology is thorough and of high quality.

There is a significant amount of data generated and presented in this analysis, so understandably there are challenges in data presentation. However, the figures are currently very difficult to read and will be even more difficult to read when formatted for a publication. Perhaps moving some of the data to the supplement might help. The font size should also be increased so that the reader does not have to zoom in on a pdf, as this will make reading on a printout almost impossible. Other than this, the manuscript provides an abundance of interesting molecular and biophysical data that will add to our understanding of neoantigen recognition in the anti-tumor T cell mediated response.

We thank the reviewer for their positive feedback. We have increased figure and font sizes throughout the paper, and have divided data into additional main figures. Further, we have rearranged our supplemental data in part in response to the reviewer's comments as well; in particular, we have moved all modeling data to the supplement, while keeping experimental structural data in the main figure section.